# Enforced MYC expression directs a distinct transcriptional state during plasma cell differentiation

Panagiota Vardaka[1], Ben Kemp[1], Sophie Stephenson[1], Eden Page[1], Matthew A Care[2,3], Michelle Umpierrez[1], Adam Annahar[1], Eleanor O'Callaghan[1], Roger Owen[4], Daniel J Hodson[5], Gina M Doody[1,3], Reuben M Tooze[1,3,4]

MYC provides a rheostat linking cell growth and division. Deregulation of MYC drives transformation in aggressive B-cell neoplasms, often accompanied by BCL2-mediated apoptotic protection. We assess how MYC and BCL2 deregulation impacts on the ability of human B cells to complete plasma cell (PC) differentiation. As B cells differentiate, MYC deregulation has little impact on the regulatory circuitry controlling B-cell identity. Induction of transcriptional regulators BLIMP1 and IRF4 remains intact and accompanies loss of B-cell surface markers. However, such differentiating cells develop an aberrant surface phenotype with reduced expression of phenotypic markers of differentiation. Although functional antibody secretion is established, enforced MYC expression dampens the expression of secretory programmes associated with PC differentiation. Accompanying this, diverse changes in the expression of genes related to translation and metabolism are observed. The establishment of this aberrant differentiated state depends on MYC homology box II. This dependence is profound and resolves to residue W135.

## Introduction

The transcription factor c-MYC (MYC) was the first oncogene deregulated by chromosomal translocation to be identified in B-cell lymphoma (1, 2, 3). It is one of the most frequently deregulated oncogenes in aggressive lymphoid cancer but also acts as a central regulator of physiological lymphocyte growth and proliferation (4, 5, 6, 7, 8). MYC acts as a sequence-specific transcription factor of the basic helix–loop–helix (bHLH) domain family, occupying E-box DNA sequence elements in complex with its obligatory partner MAX (9, 10). With high MYC expression, its genomic occupancy spreads to sites with non-consensus E-box motifs, and MYC occupancy correlates with RNA polymerase loading at primed promoters and with release of paused polymerases (9, 11, 12, 13, 14). Distinct models of MYC function support action as a global enhancer of prevailing active promoters, and as a more selective regulator of specific gene programmes that overlap between multiple cell types (9, 12, 13, 14, 15). Exit from cell cycle and cellular terminal differentiation is linked to repression of MYC and nuclear exclusion (16). In B-cell differentiation to the plasma cell (PC) stage, repression of *MYC* has been attributed to the transcription factor BLIMP1/PRDM1 (17, 18).

MYC-driven cellular transformation depends on DNA binding and on the MYC N-terminal transactivation domain (TAD) (10, 19). This TAD contains evolutionarily conserved regions, the MYC boxes (MB), that are responsible for distinct cofactor interactions (15, 20, 21). MBI contains a phosphodegron sequence controlling MYC degradation via the proteasome (22, 23). A crucial residue in MBI is T58, which is frequently mutated in aggressive lymphoma (23, 24, 25). MB0 and MBII are implicated in transactivation and transformation activity with MBII identified as essential for MYC-driven transformation (26, 27, 28). MBII mediates recruitment of TRRAP and associated histone acetyltransferase complexes (20, 26, 27, 28). At the core of MBII is a highly conserved four amino acid sequence (DCMW). W135 is highly conserved in MYC family proteins (29) and sits at the heart of the predicted MBII interface with TRRAP (30), an interface that may be therapeutically targetable (31).

MYC transforming activity is held in check by induction of apoptosis (19, 32, 33, 34, 35). Hence, *MYC* deregulation in cancers is often accompanied by *TP53* inactivation or deregulated *BCL2* (36, 37). A range of aggressive B-cell neoplasms carry oncogenic events that arrest cells during differentiation between B-cell activation and PC differentiation (38). MYC deregulation is a recurrent event in this context as a result of translocation or stabilizing mutations (1, 3, 24, 25, 39). Lymphomas with translocation of both *MYC* and *BCL2*, "double-hit lymphoma," as well as cases with MYC and BCL2 co-expression, without underlying translocation, "double-expressing lymphoma," have an adverse prognosis (39, 40). Recently, efficient transduction of primary human B cells with oncogene combinations has provided a basis for in vitro modelling of human aggressive B-cell lymphoma (41,

[1]Division of Haematology and Immunology, Leeds Institute of Medical Research, University of Leeds, Leeds, UK [2]Epidemiology and Cancer Statistics Group, Department of Health Sciences, University of York, York, UK [3]NIHR Leeds BRC, Leeds, UK [4]Haematological Malignancy Diagnostic Service, Leeds Teaching Hospitals NHS Trust, Leeds, UK [5]Welcome MRC Cambridge Stem Cell Institute, Cambridge, UK

Correspondence: r.tooze@leeds.ac.uk

42). In this context, MYC and BCL2 co-deregulation provided an example of a transforming combination driving sustained population expansion (42).

Differentiation can oppose cellular transformation by driving cell cycle exit and limiting cellular plasticity. Concomitant with this, MYC expression generally declines with differentiation (12, 43). We have developed models of human B-cell activation that are permissive for differentiation to a long-lived PC state (44, 45, 46). Driven by signals mimicking antigen receptor ligation and T-cell help, including transient CD40L exposure, B cells undergo a process of cell growth and division in which endogenous MYC expression is first induced after activation and then repressed as the differentiating cells complete cell division and transition to a specialized secretory state, recapitulating physiological PC differentiation (44). At the heart of the process of PC differentiation is a coordinated reorganization of transcription factors (47). Overall, the sequence of transcriptional regulation coordinates a MYC-associated burst of cell growth and division, with eventual repression of elements of the B-cell state and a switch to secretory gene expression (6, 14, 47).

A trigger for the transition from growth programme to PC differentiation is release from sustained CD40L signals (48, 49), which provides potent NFκB pathway activation (50), and a signal that can delay and/or prevent differentiation of activated B cells (51). Sustained provision of CD40L was integral to previous in vitro modelling of human B-cell lymphomagenesis (42). By sustaining CD40L signalling, the approach did not address whether oncogene deregulation sufficed to transform B cells under conditions permissive for PC differentiation. Examining this is of interest because it would test the impact of deregulation of MYC in the context of an intrinsically reorganizing transcriptional programme of differentiation. To address these questions, we have evaluated the impact of MYC deregulation in the context of BCL2 co-expression across human PC differentiation. Our data argue that under conditions permissive for differentiation, MYC diverts expression towards a distinct non-physiological pattern that alters surface phenotype and metabolic and growth-related gene expression. These impacts of MYC are independent of MBI but depend in part on MB0 and are dependent on MBII and the single amino acid W135.

# Results

## Acute MYC and BCL2 overexpression drives an aberrant B-cell differentiation phenotype

We aimed to test to what extent deregulated expression of MYC in combination with BCL2 impacted acutely on human B-cell differentiation. We initially evaluated a T58I variant of MYC in combination with BCL2, as this combination has been previously used in lymphoma modelling (42). By including the T58I lymphoma-associated MYC-stabilizing mutation in the MBI domain (24, 25), this approach combined overexpression and stabilization to enhance the potential for MYC impact. We tested this in the context of our differentiation system, which is permissive for human B-cell differentiation to a long-lived PC state (Fig 1A). Briefly in this model

system, B cells are activated by signals that include antigen receptor ligation, CD40 stimulation, and cytokines IL-2 and IL-21 for 3 d, during which B cells grow and begin to divide and endogenous MYC is expressed. At day 3, CD40 and antigen receptor ligation are removed and NFκB signalling is rapidly lost. Activated B cells subsequently divide rapidly while transitioning to a plasmablast state. At day 6, plasmablasts are transferred to IL-6 and APRIL or other cytokine-containing conditions that support further differentiation to the PC state (44, 45, 46). A significant difference from previous in vitro modelling of lymphomagenesis in which B cells were continuously maintained in CD40-stimulating conditions (42) is the removal of CD40 stimulation and NFκB activation at day 3 in our model supporting PC differentiation (48). A distinct pattern of NFκB activation is subsequently reintroduced upon the addition of APRIL at day 6, which supports differentiation to the PC stage (46).

In the context of this model, peripheral blood memory B cells were transduced on day 2 of activation with *MYC T58I-t2A-BCL2* retroviral vector (henceforth *T58I-t2A-BCL2*) with continued CD40 stimulation conditions for 24 h before progressing into the differentiation protocol with removal of CD40 stimulation at day 3 (Figs 1A and S1A). Transduction efficiency was high, and the sustained expression of the retroviral CD2 reporter was observed to day 20, by which time PC differentiation has been established for 10 d in differentiations in the absence of transduction (Fig 1B) (45). The overexpression of MYC and BCL2 proteins was confirmed (Fig S1B). *T58I-t2A-BCL2* was associated with a change in phenotype (Fig 1C–F). This included decreased CD27 and CD138 expression, which are hallmark features linked to PC differentiation, while also showing a decrease in CD19 expression, a hallmark B-lineage antigen whose loss of expression is a feature of malignant PCs. Across multiple time points of differentiation, *T58I-t2A-BCL2* cells showed increased cell size relative to MSCV or untransduced controls (Fig S1C and D), and *T58I-t2A-BCL2* transduction drove an increase in cell number at day 13 and day 20 of differentiation (Fig S1E). A low level of EdU incorporation, indicating continued cell cycle activity, was evident at day 21, although most of the *T58I-t2A-BCL2*–transduced population no longer incorporated EdU at this time point (Fig 1G). Transition to an antibody secretory state is the hallmark of functional PC differentiation, and establishment of functional antibody secretion was evident for both IgM and IgG in *T58I-t2A-BCL2* conditions by day 6 and sustained at day 13 (Fig 1H). Thus, MYC T58I and BCL2 overexpression from the activated B-cell stage onwards resulted in increased cell size and number and PC differentiation characterized by an aberrant phenotype but with retained capacity for antibody secretion assessed at the population level.

## Enforcing MYC overexpression drives classical target genes and alters patterns of transcription factor expression

To further understand the impact of MYC and BCL2 deregulation on differentiation, we performed a time course gene expression study in control or MYC *T58I-t2A-BCL2* conditions. Gene expression was assessed at day 0—before activation; day 3—activated B-cell stage, 24 h after transduction; day 6—plasmablast stage, 4 days after transduction; day 13—early PC stage, 11 d after transduction; and day 20—established PC stage, 18 d after transduction (MSCV control

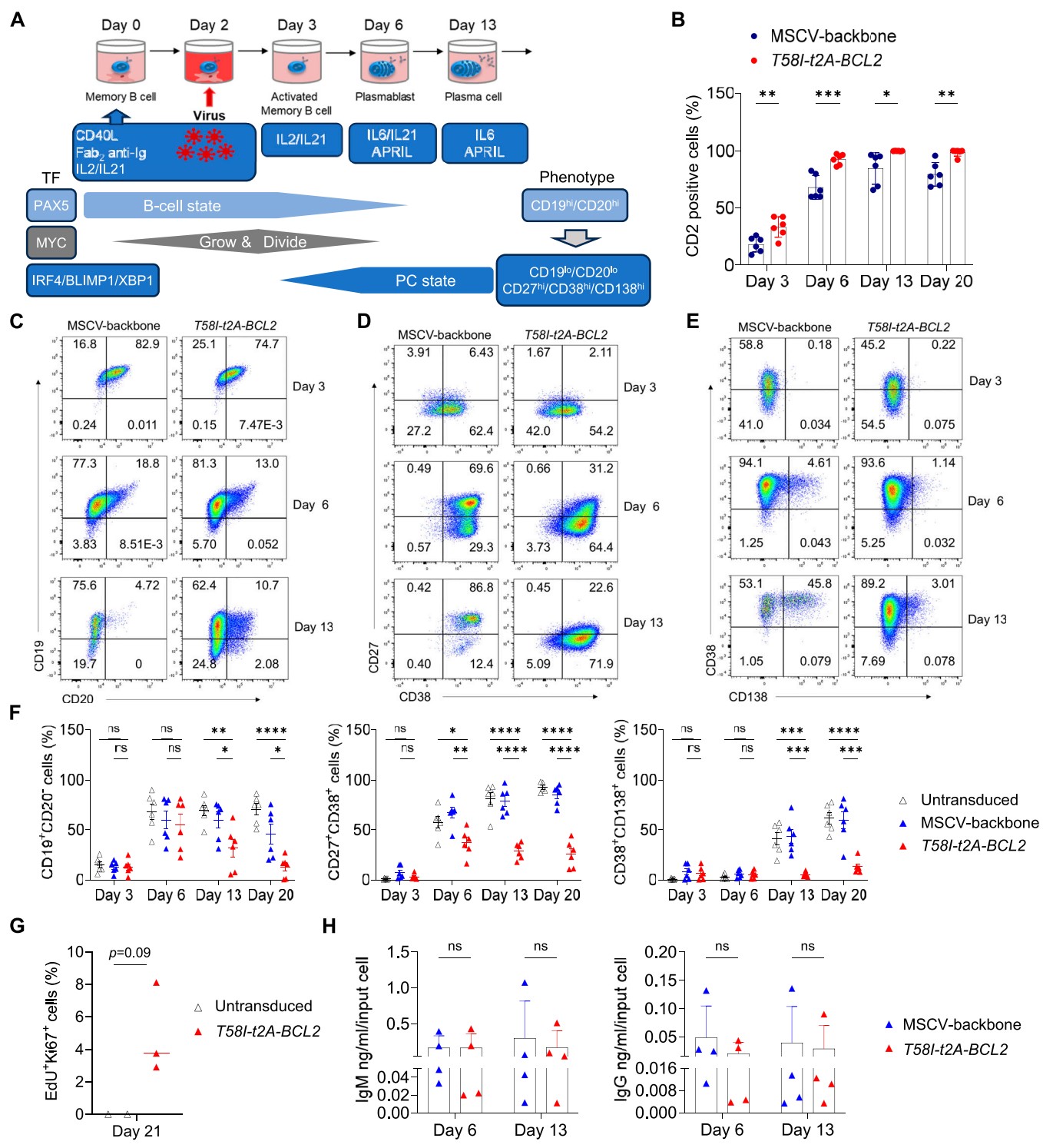

**Figure 1. Acute MYC and BCL2 overexpression drives aberrant B-cell differentiation.**
**(A)** Graphical representation of the in vitro differentiation and transduction model system. This shows general phases by day of culture at the top, followed by summary of culture conditions and biological processes below. Associated transcription factors (TF) are depicted on the left and phenotypic markers on the right. **(B)** Flow cytometry quantitation of the percentage of CD2-positive cells for MSCV-backbone and *T58I-t2A-BCL2* conditions at indicated time points. **(C, D, E)** Representative flow cytometry plots of CD19 versus CD20, CD27 versus CD38, and CD38 versus CD138, respectively, for MSCV-backbone and *T58I-t2A-BCL2* conditions at the indicated time points. **(F)** Percentages of CD19+CD20− cells (left), CD27+CD38+ cells (middle), and CD38+CD138+ cells (right) for untransduced (open triangle), MSCV (blue triangle), and *T58I-t2A-BCL2* (red triangle) samples at the indicated time points. **(G)** Percentages of EdU+Ki67+ cells at the indicated time points for untransduced (open triangle) and *T58I-t2A-BCL2* (red triangle) samples after 1-h pulse of EdU incorporation. **(C, D, E, F, G)** Data shown are pre-gated to CD2+ populations for MSCV and *T58I-t2A-BCL2*. **(H)** Quantification of total human IgM antibody (left) and IgG antibody (right) on day 6 and day 13 for the indicated conditions. Data are representative of at least two independent experiments. **(F)** Bars and error represent the mean and SD; (F) mean and SEM. **(B, F, G, H)** Unpaired two-tailed *t* test: ns, not significant; *P < 0.05; **P < 0.01; ***P < 0.001; ****P < 0.0001.

conditions generated insufficient cells for analysis at day 20) (Tables S1 and S2). Consistent with a progressive impact of MYC *T58I-t2a-BCL2* conditions on gene expression, Uniform Manifold Approximation and Projection showed similar clustering of samples at day 3 with subsequent increased separation of MYC *T58I-t2a-BCL2* conditions from controls at day 6, day 13, and day 20 (Fig 2A). *MYC* was substantially expressed in physiological differentiation at day 3 in activated B cells and was then progressively repressed in control differentiation conditions. In contrast, MYC *T58I-t2a-BCL2* conditions showed a modest increase in *MYC* expression at day 3 and then maintained supra-physiological levels of *MYC* expression throughout subsequent differentiation (Fig 2B). Expression of *CD2* and enhanced and sustained expression of *BCL2* were also confirmed in transduced conditions (Figs 2B and S2A).

We next assessed the extent to which phenotypic changes were recapitulated in gene expression data. We observed significant parallels with suppression of *SDC1* (CD138), *CD27*, and *CD19* expression at later time points in MYC *T58I-t2a-BCL2* conditions relative to controls. *CD38* expression was not substantially impacted, whereas *MS4A1* (CD20) expression was increased in MYC *T58I-t2a-BCL2* conditions (Fig 2C). These conditions were also associated with suppressed *TNFRSF17* (BCMA) but not *TNFRSF13B* (TACI), and *CD79A* but not *CD79B* expression (Fig S2B). Thus, the perturbed phenotype observed by flow cytometry was reflected in corresponding changes at the transcript level.

PC differentiation is driven by coordinated changes in transcription factor expression (47). We therefore examined how known regulators of the B-cell state and PC differentiation were expressed (Fig 2D). *PAX5* and *EBF1*, transcriptional regulators involved in maintaining B-cell states (52, 53, 54, 55), were expressed in resting and activated B cells and then equivalently repressed during differentiation in control and MYC *T58I-t2a-BCL2* conditions. Positive regulators of PC differentiation *IRF4* and *PRDM1* (BLIMP1) were induced upon activation and differentiation with modest reductions observed in maximal expression for both factors in MYC *T58I-t2a-BCL2* conditions. *XBP1*, the primary transcriptional driver of secretory reprogramming and the unfolded protein response (UPR) of the endoplasmic reticulum (ER) (56, 57, 58), showed suppressed induction at day 6 and all subsequent time points in MYC *T58I-t2a-BCL2* conditions, but nevertheless retained elevated expression relative to days 0 and 3. Differential regulation of other transcription factors was also observed including repression of *RUNX1*, which has a role in cell cycle entry (59), and enhanced expression of the PC fate antagonist *BACH2* (60, 61, 62) and *SREBF1*, a controller of sterol metabolic pathways (63), in MYC *T58I-t2a-BCL2* conditions (Fig S2C).

Consistent with canonical MYC-driven gene expression, well-defined MYC targets such as *TERT*, the catalytic component of telomerase (64), *JAG2*, a receptor on the NOTCH signalling pathway (65), *TRAP1*, a key mitochondrial chaperone (66), and *FABP5*, linked to fatty acid metabolism and a potential therapeutic vulnerability in myeloma (67), were increased (Fig 2E). A wide range of other MYC target genes defined in cellular models in both mouse and human (6) were also profoundly induced in MYC *T58I-t2a-BCL2* conditions (Tables S1 and S3).

Because XBP1 is a principal regulator of secretory reprogramming, we assessed known XBP1 and ER stress response target

genes. We found consistent but modest patterns of dampened expression for genes such as *HERPUD1*, *ERLEC1*, *DERL3*, and *TXNDC5* (68, 69, 70), which share direct XBP1 promoter occupancy at the plasmablast stage in our model system (Fig 2F) (48). Immunoglobulin is the main secretory output of PCs, and expression is profoundly decreased on conditional *XBP1* deletion in murine PCs (71, 72). We therefore also examined the expression levels of immunoglobulin genes. Indeed, these genes comprised some of the most differentially expressed genes and showed significantly dampened expression in MYC *T58I-t2a-BCL2* conditions at later time points, particularly for *IGHG1*, *IGHG2*, *IGHG3*, and *IGHM* (Fig S2D). Nonetheless in all instances, including in the context of MYC *T58I-t2a-BCL2* conditions, the expression at day 6 and beyond was higher than that observed at earlier time points for both secretory pathway and immunoglobulin genes. Thus, analysis at the individual gene level suggested a coordinated impact of MYC overexpression on the regulation of secretory pathways during PC differentiation with a dampening rather than amplification of gene expression related to antibody secretory state.

## MYC and BCL2 overexpression drives coordinated modular patterns of gene expression change

To test gene regulation at a global level, we analysed gene expression changes in MYC *T58I-t2a-BCL2* and control conditions using Parsimonious Gene Correlation Network Analysis (PGCNA) (73). This correlation-based method provides an alternate form of dimensionality reduction and allows the shifting patterns of gene expression across differentiation and between conditions to be assessed in terms of modules of coregulated genes. PGCNA identified 16 modules of coregulated genes (labelled M1–M16 according to the number of module genes) (Table S3) with distinct patterns of expression across the differentiation and between control and MYC *T58I-t2a-BCL2* conditions (Fig 3). Gene ontology and signature enrichment analysis were used to assess features of known biology associated with each of these coregulated gene modules (Table S4).

The modules resolved coregulated genes relating to B-cell differentiation, cell cycle, MYC function, translation, and metabolism (Fig 3 and Table S4). Four primary patterns were identified, these were: (1) modules repressed on differentiation; (2) modules sustained or induced in expression with MYC *T58I-t2a-BCL2*; (3) modules sustained or induced on differentiation but repressed with MYC *T58I-t2a-BCL2*; and (4) modules induced on differentiation and enhanced with MYC *T58I-t2a-BCL2*.

For modules following the general kinetics of "repressed on differentiation," the enrichments of signature terms included features of the B-cell state including targets repressed by BLIMP1 (M1), genes regulated by NFκB and other signalling pathways (M2), and cell cycle (M11). Of these, cell cycle–related gene expression retained a higher level of expression at day 6 and day 13 in MYC *T58I-t2a-BCL2*, albeit at levels substantially lower than the day 3 peak, and was ultimately repressed by day 20.

For modules with the general kinetics of "sustained or induced with MYC," the associated genes were enriched for classical MYC targets (MSigDB_Hallmark_MYC_Targets_v1 and v2) (M4 and M6) and MTORC1 signalling (M4) and mitochondrial components (M6)

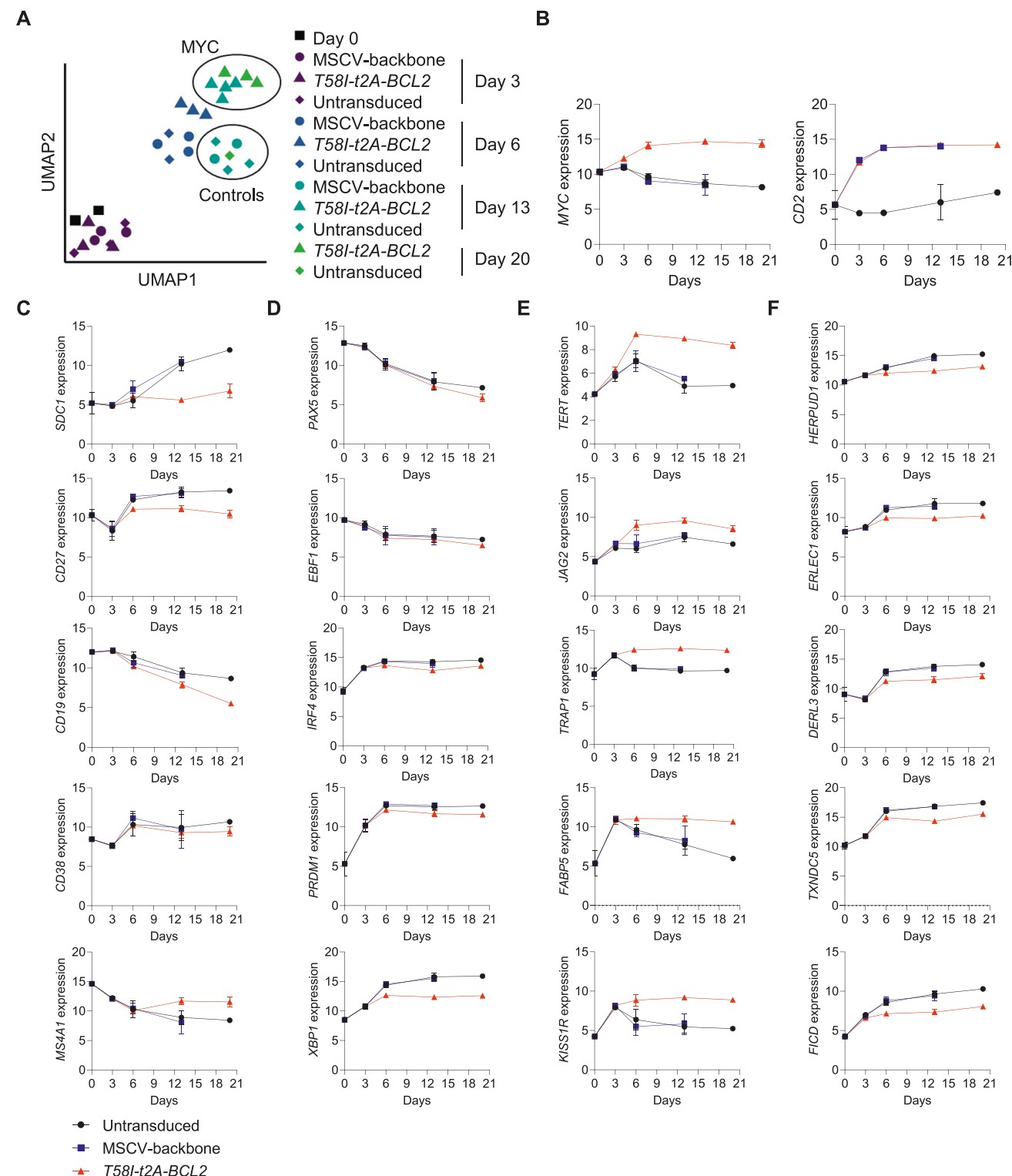

**Figure 2. MYC T58I selectively perturbs gene expression in B-cell differentiation.**
**(A)** Uniform Manifold Approximation and Projection of differentially expressed genes for the indicated conditions and time points. **(B, C, D, E, F)** Log$_2$-normalized RNA-seq expression values (y-axis) of selected genes across the differentiation time course (x-axis) as indicated for untransduced, MSCV, and *T58I-t2A-BCL2* conditions (bottom left). **(B, C, D, E, F)** Gene expression is shown as indicated in the figure for (B) *MYC* and *CD2*; (C) surface proteins linked to immunophenotyping; (D) transcription factors; (E) known MYC targets; and (F) XBP1 targets. Data are representative of two independent experiments with a total of n = 1–4 samples per time point and condition. Bars and error represent the mean and SD. FDR-corrected *P*-values for all pairwise conditions at each time point are provided separately in Table S2.

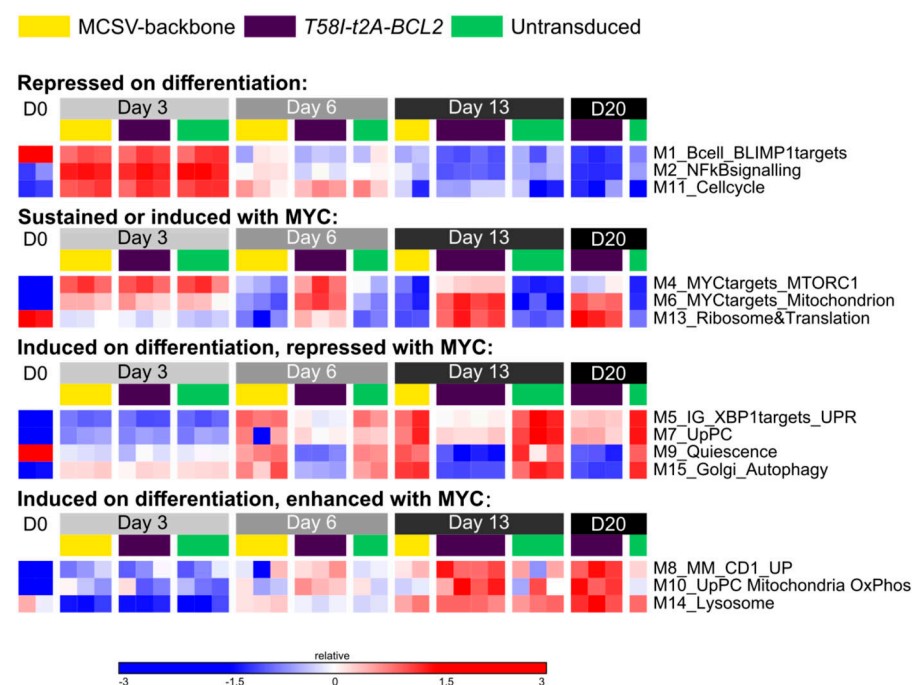

**Figure 3. Expression network-level analysis of MYC T58I on B-cell differentiation.**
Parsimonious Gene Correlation Network Analysis was used to define modules of coregulated genes. Heat map of module-level gene expression with expression patterns represented as the median z-score of the 10 most connected genes per module scale (–3 blue to +3 red). Modules are divided into four patterns of expression change as indicated above each grouping. Module number and indicative summary terms of associated ontologies are shown on the right. Time points are indicated in the grey to black bars at the top of the figure going from day 0 (D0, left) to day 20 (D20, right). Individual conditions are identified using the code as indicated at the top of the figure: yellow (MSCV), purple (*T58I-t2A-BCL2*), green (untransduced). Data are representative of two independent experiments with a total of n = 1–4 samples per time point and condition.

and ribosome and translation genes (M13). Of note, the kinetics of these modules varied such that M4 was induced in activated B cells and sustained in expression in the presence of MYC but gradually repressed on further differentiation, whereas M6 and M13 showed sustained induction throughout the differentiation. This group of modules confirmed an expected impact of MYC on previously defined hallmark target genes, while in addition indicating that the regulation amongst these genes is not uniform across the course of differentiation.

Modules with the general kinetics of "induced on differentiation, repressed with MYC" contained genes enriched for a range of features associated with immunoglobulin genes, XBP1 targets and the UPR (M5), other features of the PC state (M7), transcription factors and genes associated with quiescence in haematopoietic cells (M9), and the Golgi apparatus and autophagy (M15).

Modules following the general kinetics of "Induced on differentiation, enhanced with MYC" contained the associated genes enriched for features related to molecularly defined subsets of plasma cell myeloma and glycine/serine metabolism (M8), and genes up-regulated in antibody-secreting cells including mitochondrial oxidative phosphorylation (M10).

This modular analysis therefore confirmed the broad impact of enforced MYC expression across differentiation including an expected distinct signal for enhanced MYC target gene expression. Across the differentiation time course, the overall impact of overexpressed MYC *T58I-t2A-BCL2* was not fixed, nor did it reflect an amplification of the physiological differentiation programmes associated with plasma cell differentiation. Instead, the impact of MYC developed across the course of the differentiation towards a distinct aberrant expression state in which suppression of features of the B-cell state was linked to dampened expression of secretory pathway and plasma cell features and enhanced expression of a subset of translation and metabolism-related expression programmes.

## MB domains of MYC show differential contributions to gene regulation during PC differentiation

The MYC TAD has been implicated as critical in transforming activity in various cellular models; we therefore next aimed to test the contribution that MB0, MBI, or MBII domains of the MYC TAD made to the divergent programming of perturbed PC differentiation. To this end, we generated a MYC *WT-t2A-BCL2* vector (i.e., with T58 not T58I) and vectors in which MB0, MBI, or MBII was deleted in this context (Fig S3A). We obtained similar transduction efficiencies to our previous results (Fig S3B). The overexpression of MYC and BCL2 proteins in comparison with the MSCV-backbone control was validated for all three MYC MB deletion mutants tested (Fig S3C and D). Despite the overexpression of MYC, BLIMP1, the key transcriptional regulator linked to PC fate determination, was present at similar levels in all conditions. Relatively enhanced MYC expression was observed for *ΔMBI-t2A-BCL2*, which deletes the phosphodegron sequence. In contrast, *ΔMB0-t2A-BCL2* and *ΔMBII-t2A-BCL2* showed relatively reduced levels of expression when compared to MYC WT at day 6. Because BCL2 expression levels were similar across the transductions, the differences in MYC expression associated with the MB deletions may reflect differences in protein stability. Nonetheless, exogenous MYC was expressed at higher levels than in control conditions.

MYC *WT-t2A-BCL2* recapitulated the phenotypic effect observed for MYC T58I, and similarly, *ΔMBI-t2A-BCL2* showed little difference in phenotype from MYC *WT-t2A-BCL2* (Figs 4A and B and S3E). In contrast, *ΔMB0-t2A-BCL2* and *ΔMBII-t2A-BCL2* diverged from the MYC-associated phenotype and differentiations reverted towards

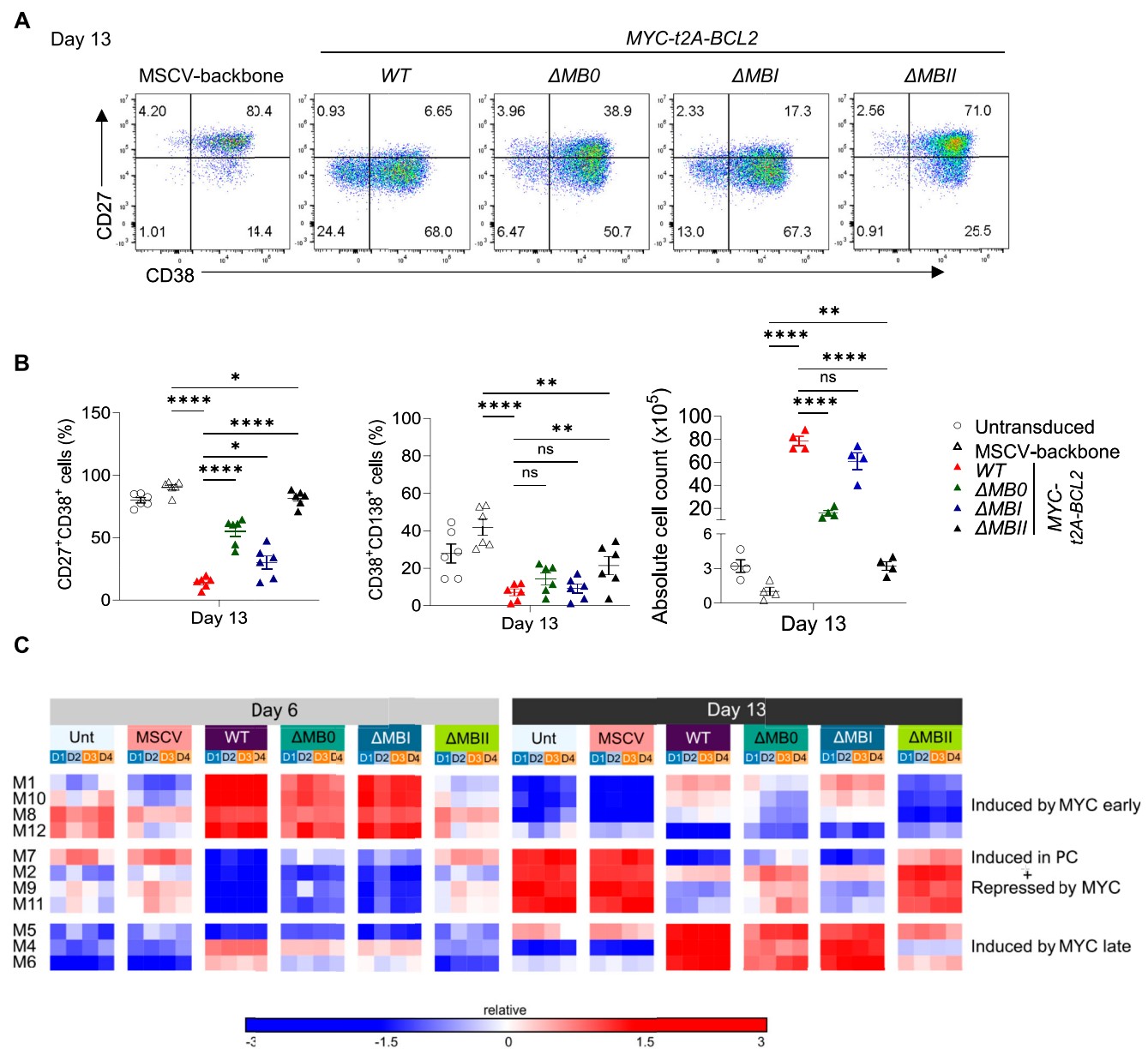

**Figure 4.** **Deletion of MYC TAD MB0, MBI, and MBII has differential effects on MYC-driven phenotypic and expression features.**
**(A)** Representative flow cytometry plots at day 13 for control MSCV-backbone, MYCwt, ΔMB0, ΔMBI, and ΔMBII conditions as shown for CD27 versus CD38. **(B)** Summary of flow cytometrically defined percentages of CD27⁺CD38⁺ cells (left) and CD38⁺CD138⁺ cells (middle) and haemocytometer-derived absolute cell number (right) at day 13 for the indicated conditions. **(A, B)** Data are representative of at least two independent experiments. Flow cytometry data shown for the transduced conditions are pre-gated to CD2⁺ populations. **(B)** Bars and error represent the mean and SEM. Unpaired two-tailed $t$ test: ns, not significant; *$P < 0.05$; **$P < 0.01$; ****$P < 0.0001$. **(C)** Parsimonious Gene Correlation Network Analysis defined modules of coregulated genes shown as a heat map of module-level gene expression with expression patterns represented as the median z-score of the 10 most connected genes per module scale (−3 blue to +3 red). Modules (numbered on left) are clustered according to the pattern of regulation in response to enforced MYC expression (right). Time points are indicated as grey at day 6 and black at day 13. Individual control or transduction conditions are indicated with the colour code identified in the figure and samples from different donors illustrated in the blue to orange colour code beneath. **(C)** Data are representative of two independent experiments with a total of n = 4 samples per time point and condition.

phenotypic patterns of control conditions. In terms of absolute cell count, MYC *WT-t2A-BCL2* enhanced cell number at day 13 with no significant difference for *ΔMBI-t2A-BCL2*; however, both *ΔMB0-t2A-BCL2* and *ΔMBII-t2A-BCL2* showed significantly reduced impact on cell number relative to *WT-t2A-BCL2* (Fig 4B, right panel). Indeed, *ΔMBII-t2A-BCL2* retained only a marginal increase relative to MSCV

but did not differ significantly from untransduced control conditions in terms of cell numbers.

To further assess the impact of MB deletions, we studied gene expression at day 6 (plasmablast) and day 13 (PC stage) (Table S5). Multidimensional scaling (MDS) showed that expression separated both in terms of differentiation state—separating

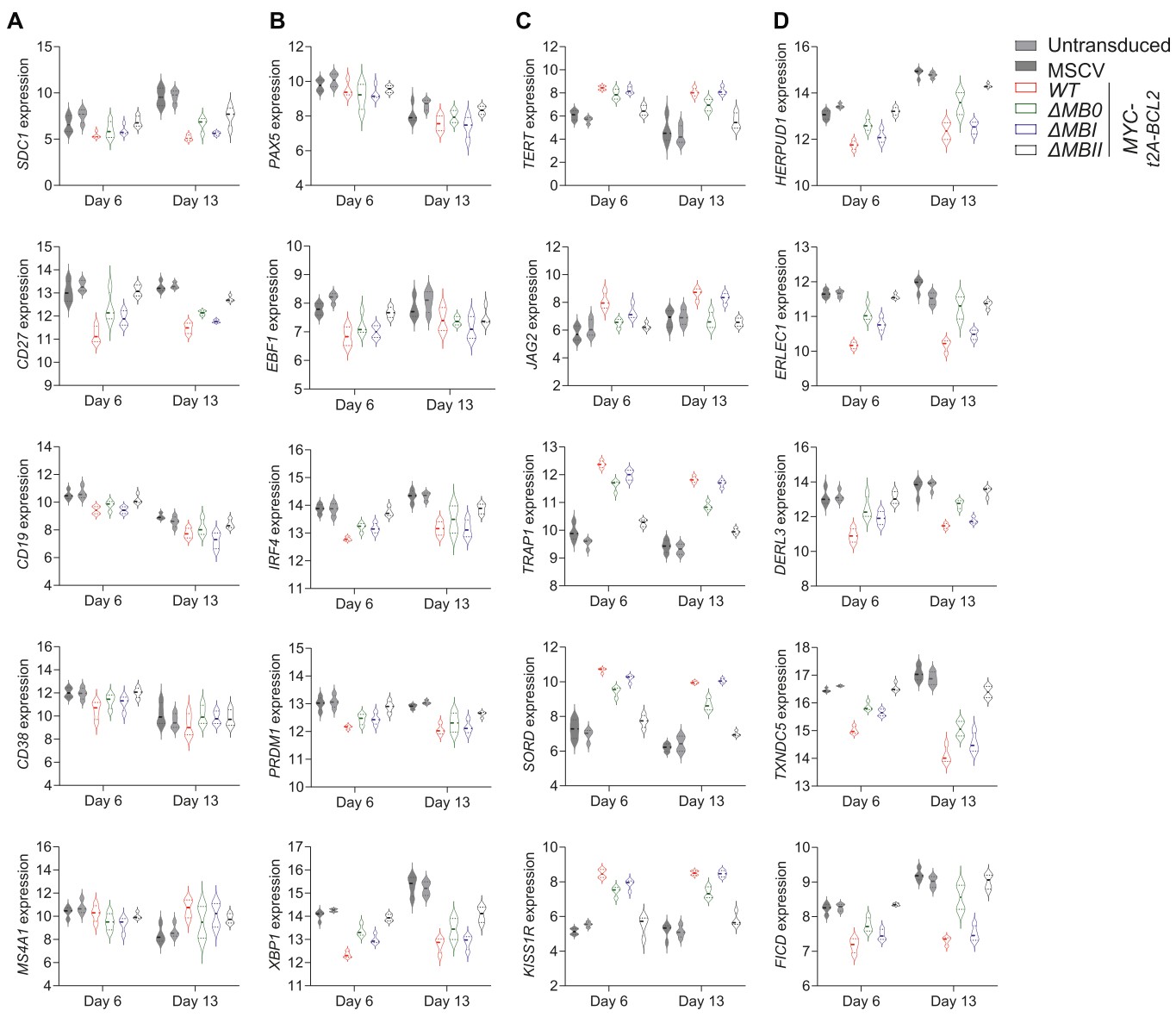

**Figure 5. MB0, MBI, and MBII deletion impacts on indicative gene regulation and on functional secretory output.**
Violin plots of log$_2$-normalized RNA-seq expression values of individual genes plotted at day 6 (left side of graphs) and day 13 time points (right side of graphs) for the indicated conditions (top right of the figure). **(A, B, C, D)** Genes shown are indicated to the left of each graph for (A) surface antigens; (B) transcription factors; (C) MYC targets; and (D) XBP1 targets. Data are representative of two independent experiments with a total of n = 4 samples per time point and condition. FDR-corrected *P*-values for all pairwise conditions at each time point are provided separately in Table S2.

samples at day 6 from those at day 13—and according to MYC transduction at each time point (Fig S3F). This was consistent with the hierarchical order of phenotypic impact such that MYC *WT-t2A-BCL2* was most and *ΔMBII* was least distinct from controls at both day 6 and day 13. We explored coordinated patterns of gene regulation with PGCNA resolving modules of genes, which showed distinct kinetics between day 6 and day 13 and differential impact by MYC and MB deletions (Fig 4C). Broadly, these were divided into patterns: (1) induced by MYC early (day 6 > day 13, e.g., M1 and M10 enriched for Hallmark_MYC_Targets_v1/v2 and M8 enriched for Hallmark_E2F_Targets and G2M_Checkpoint); (2) induced in PCs and repressed by MYC (e.g.,

M7 enriched for Golgi and endomembrane system genes, M9 Immunoglobulin genes and PC features); and (3) induced by MYC late (day 6 < day 13, e.g., M4 enriched for ribosome components and translation elongation) (Fig 4C and Tables S6 and S7). MYCwt, *ΔMB0*, and *ΔMBI* showed similar patterns of module regulation that diverged from those observed in controls. *ΔMB0* showed reduced intensity of effect for both MYC up- and down-regulated modules. In contrast, *ΔMBII* diverged from the other MYC conditions with a pattern of modular gene expression resembling that of the control conditions (Fig 4C). Therefore, in the context of both dimensionality reduction using MDS and modular expression analysis using PGCNA, the MB deletion

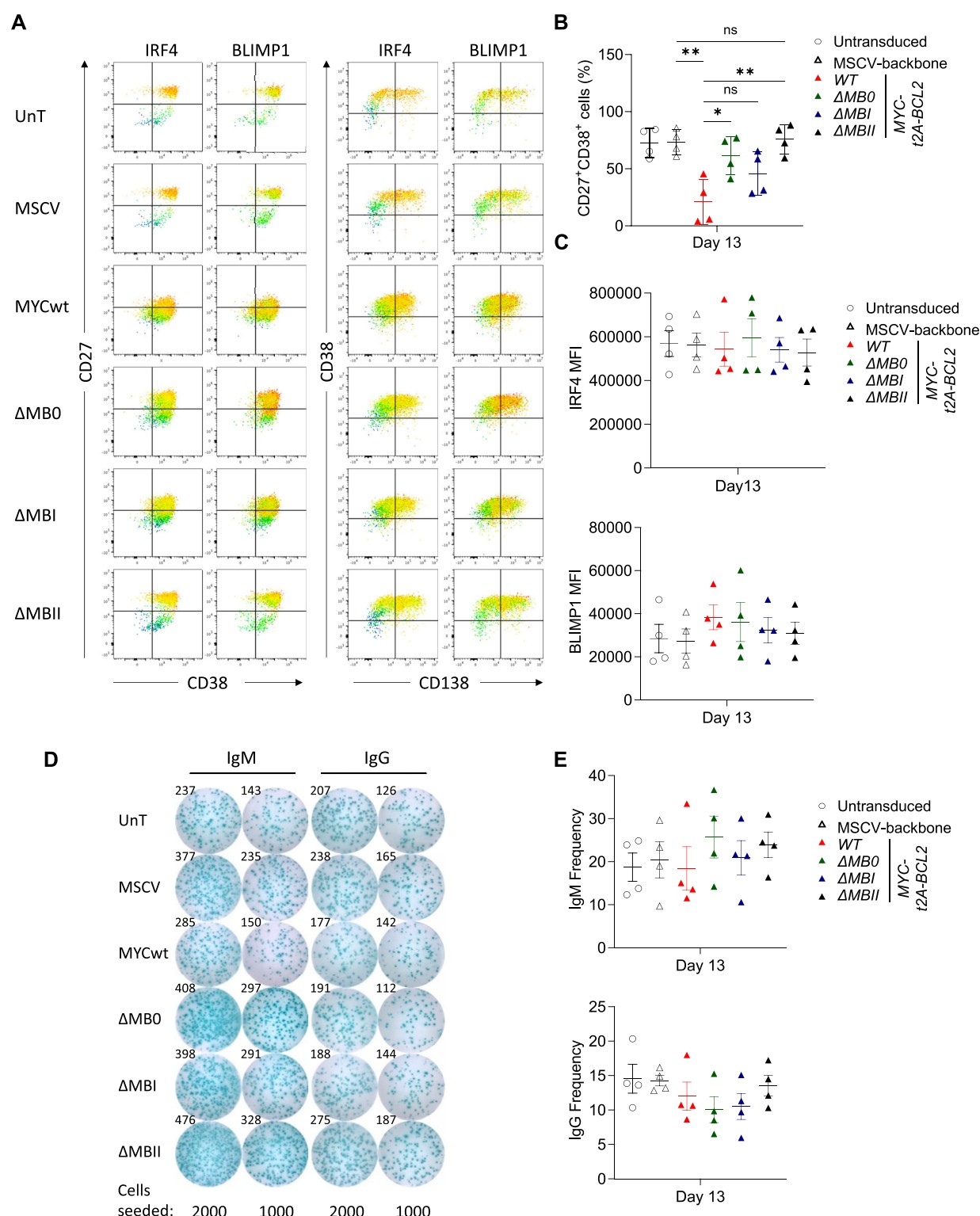

**Figure 6. MYCwt and MB deletions do not prevent the formation of functional ASCs.**
(A) Representative flow cytometry plots at day 13 for conditions identified (left of row) for intracellular IRF4 (left pair) and BLIMP1 (right pair)—intracellular antigen MFI plotted as gradient colour scale (blue [minimum] to red [maximum]) projected onto CD27 versus CD38 (left) and CD38 versus CD138 (right). (B) Flow cytometry data summary of CD27⁺CD38⁺ cell percentages at day 13 for indicated conditions. (C) IRF4 MFI (top) and BLIMP1 MFI (bottom) for indicated conditions at day 13. (A, B, C) Data are derived from four independent donors and are pre-gated to CD2⁺ populations except for the untransduced condition. (D) Representative ELISpot data for IgM (left) and IgG (right) across conditions indicated and cell seeding densities (below) at day 13. (E) Estimated ASCs per 100 cells seeded for IgM (upper panel) and IgG (lower panel) at day 13 for four donors and indicated conditions. Bars and error represent the mean and SD. (B) Unpaired two-tailed *t* test: ns, not significant, *P < 0.05; **P < 0.01.

mutants followed a hierarchy of impact on the consequences of MYC overexpression: ΔMBII > ΔMB0 > ΔMBI ≈ MYCwt.

To consider this at a gene level, we revisited the expression patterns of index genes linked to B-cell differentiation, MYC impact, and XBP1 function. Here, we observed that although *MYC* and *CD2* expression levels were similar in all conditions (Fig S4A and B), a hierarchy was evident across both MYC-responsive and MYC-repressed features at the individual gene level for surface proteins, transcription factors, MYC targets, UPR/ER genes, and immunoglobulin genes (Figs 5 and S4C).

### Differentiation to an antibody-secreting cell state is retained with the enforced expression of MYC

Initial ELISA data indicated that antibody secretion was established in differentiations despite enforced MYC expression. This contrasted with a decrease in gene expression related to secretory reprogramming observed under the same conditions. We therefore sought to address whether there were differences in an antibody-secreting cell (ASC) state at the cellular level. Differentiations with *WT-t2A-BCL2* and MB deletion mutants were repeated and monitored for phenotypic changes, intracellular expression of transcription factors, and antibody secretion using ELISpot. Flow cytometry confirmed that both IRF4 and BLIMP1 were similarly induced in the context of MYC overexpression as B cells differentiated (Fig 6A–C). XBP1 intracellular staining was technically challenging and could not be adequately assessed (high background with isotype control, no difference between conditions). Consistent suppression of CD27⁺/CD38⁺ and CD38⁺/CD138⁺ phenotypes was observed with MYCwt and ΔMBI conditions; however, the intensity of IRF4 and BLIMP1 expression remained correlated with phenotypic markers of the PC state within the aberrant phenotypic state (Fig 6A). Thus, the aberrant phenotype observed upon enforced MYC expression was not associated with a reduction in BLIMP1 or IRF4 protein expression. The frequency of ASCs was assessed by ELISpot for IgM and IgG at day 13 of differentiation (Fig 6D and E) using cells from the same differentiations. ELISpot demonstrated robust induction of ASC activity across all conditions. There was no significant difference in ASC number for either IgM or IgG.

Therefore, despite the dampening of PC- and UPR-associated gene expression observed at the cell population level after enforced MYC overexpression, these effects did not translate into an observable difference on the ASC state. We conclude that MYC overexpression and the associated aberrant phenotypic and expression state were permissive for ASC differentiation.

### The DCMW motif and W135 are critical for the effect of MYC MBII on human PC differentiation

Given the apparent dependence on MBII, we next addressed to what extent this could be attributed to the core conserved sequence of MBII: the DCMW motif (aa132–135) or W135 alone, the most highly conserved residue in MBII, which sits at the heart of the predicted TRRAP interaction (30). Substitutions DCMW/AAAA or W135A were generated in the context of the *MYC-t2A-BCL2* configuration (Fig S5A), and comparably, high transduction efficiency

was verified for the MBII mutants (Fig S5B). MYC, BCL2, and BLIMP1 protein overexpression was validated (Fig S5C and D). BCL2 and BLIMP1 expression was equivalent between conditions. MYC was substantially overexpressed relative to controls in all conditions but was higher for MYCwt relative to ΔMBII or MBII point mutants, again suggesting that MBII may contribute to stability (Fig S5D). Nonetheless, ΔMBII and MBII point mutants remained substantially overexpressed relative to control conditions.

The MYC *MBII-4aa* mutant or the *MBII-W135A* again significantly impacted on the phenotypic changes induced by enforced MYC expression, such that MYC *MBII-4aa−* or *MBII-W135A*–expressing cells more closely resembled control differentiations than MYCwt conditions in terms of phenotype and cell number (Figs 7A–C and S5E). Our previous analyses indicated that phenotypic effects of MYC conditions were closely related to the extent of gene expression change in the model. Indeed, in multidimensionality scaling of day 13 gene expression data ΔMBII and the two MBII mutants were clustered together with untransduced controls and separated from MYCwt conditions (Fig 7D). Absolute numbers of significantly differentially expressed genes even at lenient fold-change thresholds were profoundly reduced for ΔMBII and the two MBII mutant conditions (Tables S2 and S8, sheet D13_Fig8_SF6). PGCNA confirmed that at the modular level, MYCwt conditions again differed profoundly from controls (Fig 7E) with multiple modules identified which broadly divided into those repressed or induced by MYCwt (Tables S9 and S10). The MBII mutant conditions resembled control conditions more than MYCwt and showed both impaired induction and repression of gene modules that were responsive to MYC (Fig 7E). No significantly differentially expressed genes were observed between ΔMBII and either MBII mutant or between the two MBII mutant conditions in direct comparison. When comparing ΔMBII or either of the two MBII point mutant conditions to control conditions, very few genes remained differentially expressed; amongst these, the most consistent genes across multiple comparisons, *DEPTOR*, *EEF1A1*, *KISS1R*, *BCL2L11 (BIM)*, *EEF2*, *EVI2A*, *GAS5*, *RPL5*, *TGFBI*, and *ZNF581*, were up-regulated, whereas *XBP1* remained significantly reduced in the expression level (Figs 8 and S6A–C and Tables S2 and S7). By comparison, MYCwt drove a total of 2,582 differentially expressed genes against control conditions (Table S2). Thus, the residual component of impact that was consistent across the ΔMBII and both MBII mutant conditions amounted to 0.4% of the total MYCwt-induced gene expression changes at day 13 of the differentiation. We conclude that either a four amino acid DCMW/AAAA substitution or a single W135A substitution phenocopied deletion of MBII. The single amino acid substitution W135A suffices to abrogate the ability of enforced MYC expression to drive aberrant gene expression and associated phenotypic changes during plasma cell differentiation.

## Discussion

Deregulation of MYC is one of the key drivers of aggressive B-cell malignancies (4). At the same time, precise control of MYC expression is essential to coordinate cell growth and proliferation

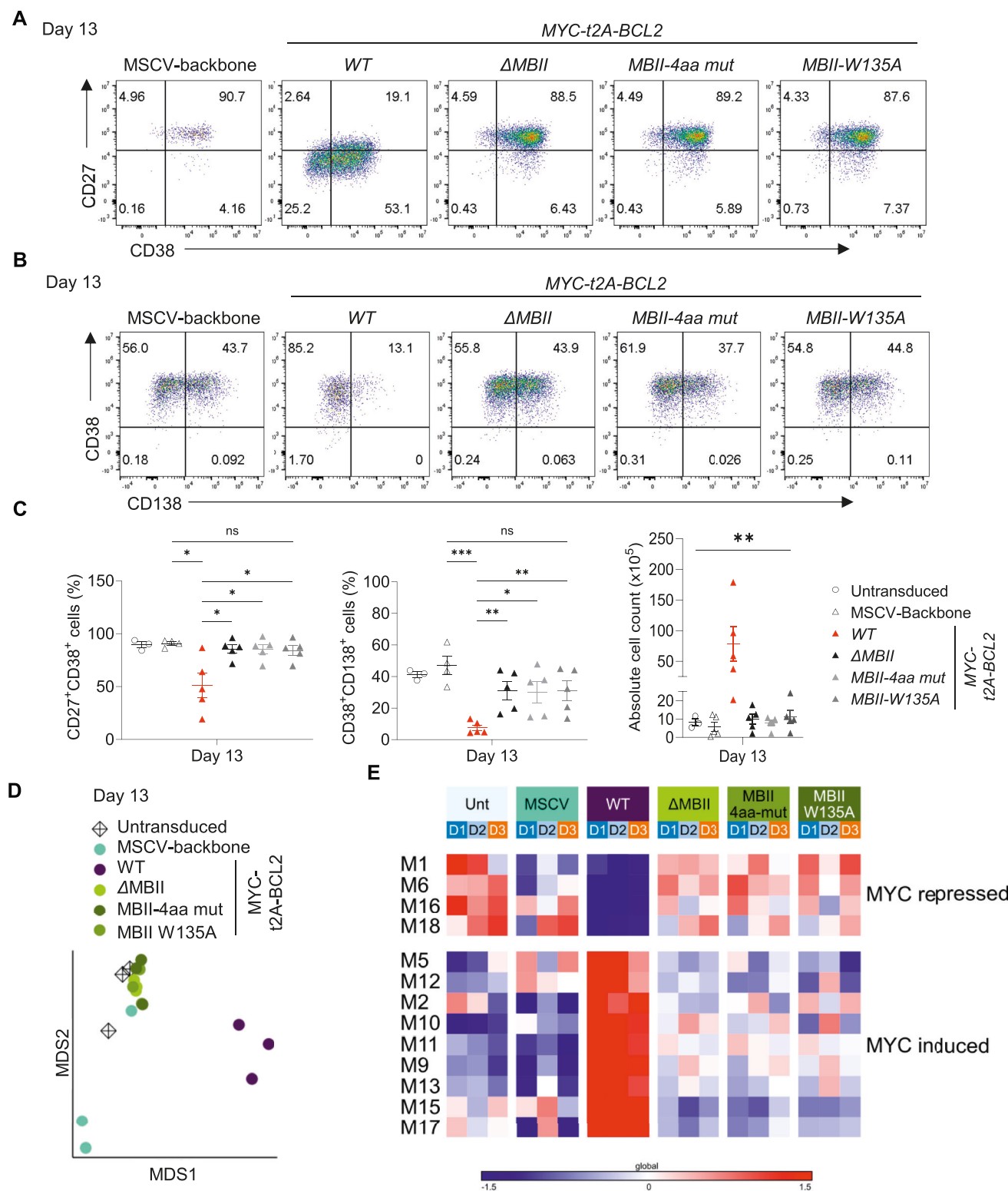

**Figure 7. Point mutation of the DCMW motif and W135 phenocopies MBII deletion.**
**(A, B)** Representative flow cytometry plots at day 13 for the indicated conditions above each dot plot for (A) CD27 versus CD38; and (B) CD38 versus CD138. **(C)** Flow cytometry quantification of the percentage of CD27+CD38+ cells (left), CD38+CD138+ cells (middle), and haemocytometer-derived cell count (right) at day 13 for the conditions indicated to the right of graphs. **(A, B, C)** Data are representative of at least three independent experiments. Data shown, apart from the untransduced condition, are pre-gated to CD2+ populations. **(C)** Bars and error represent the mean and SEM. Unpaired two-tailed *t* test (left and middle graphs); one-way ANOVA (right

with differentiation in functional lymphocyte expansion of the immune response (5, 6). Here, we have addressed the acute impact of MYC overexpression during human B-cell differentiation to the PC stage. We have assessed the extent to which MYC overexpression prevents or perturbs PC differentiation and the extent to which the impact of MYC overexpression changes as the cellular context progresses along the differentiation trajectory. We tested this both for MYC carrying the T58I mutation found in aggressive lymphoma, which stabilizes MYC expression, and for WT MYC lacking this mutation (25). MYC deregulation drove gene expression changes consistent with previous models of lymphoma and other cancer types (6). However, the effect of MYC overexpression changed as the underlying cellular differentiation state shifted towards the PC state.

Distinct models of gene regulation by overexpressed MYC have been proposed, suggesting action either as a global transcriptional activator or as a more specific regulator of distinct biological pathways (9, 12, 13, 14). In our model, MYC expression is sustained at high levels from the activated B-cell stage onwards and correlates with progressively lower endogenous MYC levels in control conditions. Our experiments included BCL2 co-expression to provide rescue from potential MYC-driven apoptosis (36, 37) and did not directly distinguish the potential effects of BCL2 overexpression alone. However, the various MYC mutants demonstrated that the gene expression changes depended on elements of MYC and the impact of BCL2 on gene expression was negligible. We can therefore assign the distinct expression states to MYC activity with reasonable confidence. Genes that were enhanced or suppressed by overexpressed MYC during differentiation were significantly linked to known biology and enriched for hallmark MYC target genes and those with classical E-box motifs. The features linked to MYC overexpression evolved over the course of differentiation and did not reflect enhanced expression of the prevailing patterns of gene expression linked to stages of activated B cell to plasma cell differentiation. Rather, overexpressed MYC established a distinct aberrant expression state during PC differentiation.

Physiologically, MYC has been identified as a rheostat linking the extent of B-cell activation to cell growth and the subsequent capacity for sequential cell division (5, 6, 7). In this setting, MYC sits at the heart of a transcriptional circuitry, which coordinates a burst of cell growth and division with eventual differentiation and secretory reprogramming (47). During this sequence of events, external signals drive B-cell activation and expression of MYC (6, 74). These signals also induce IRF4, which in turn cooperates with STAT3 to drive the expression of PRDM1/BLIMP1 (75, 76). Accumulating expression of BLIMP1 ultimately leads to repression of MYC (17), curtailing the proliferative burst. At the same time, BLIMP1 suppresses the B-cell identity transcription factor PAX5 and other features linked to the mature B-cell state (52, 53). Loss of PAX5 releases XBP1 from PAX5-mediated repression (53, 77, 78).

Acting together BLIMP1 and XBP1 coordinates the high-level expression of immunoglobulin genes, the transition from membrane to secreted immunoglobulin encoding transcripts, and the expansion of the secretory apparatus (56, 57, 58, 71, 72, 78, 79). The suppression of MYC and the expression of negative cell cycle regulators coordinate the secretory transition to cell cycle exit (18, 80, 81). Additional transcription factors such as BACH2 and BCL6 can add further levels of control to delay differentiation and facilitate class switching or interpose the complex biology of the germinal centre response (60, 61, 62, 82, 83).

The enforced expression of MYC that we have modelled occurs at the critical juncture when the burst of physiological MYC expression is at its peak and then begins to be curtailed by the underlying reorganizing transcription factor network (48). Here, deregulated MYC expression sustains gene expression linked to the cell growth and metabolic programmes. Importantly, deregulated MYC expression has little impact on the expression of IRF4 or BLIMP1 or the transcriptional regulators of B-cell identity PAX5 or EBF1. Although we only assessed the latter at the gene expression level, the coordinated repression of features of the B-cell state including CD19 is consistent with retained functional repression of these transcription factors. Hence, the regulatory circuitry controlling changes in B-cell identity–related genes is at most marginally affected (52, 53, 54, 55, 84). Although MYC deregulation appears to dampen the expression of genes linked to secretory reprogramming, we found that this does not result in a detectable reduction in ASC generation. MYC has previously been identified as capable of binding to the XBP1 promoter and positively regulating expression in cancer models (85, 86). Indeed, MYC is often deregulated in aggressive PC malignancies in which XBP1 is expressed (87). In the context of plasma cell differentiation, our data indicate that enforced MYC overexpression is dampened rather than enhanced XBP1 expression and genes are linked to the secretory pathway and UPR, suggesting that in this context overexpressed MYC may act as a repressor at the XBP1 promoter. That the dampening of immunoglobulin and secretory pathway–associated gene expression did not result in a significant reduction in ASC generation as assessed by ELISpot might be explained by a functional overshoot in normal differentiation. The transcriptional circuitry controlling PC differentiation can be viewed as two interconnected feedforward loops. One of these regulates the pulse of growth and proliferation and the other the delayed transition to a secretory fate. In this context, the deregulated overexpression of MYC in activated B cells does not override either of the circuits. Instead, MYC overexpression shifts the differentiating B cell to an expression state, which is permissive for both loss of B-cell state and the transition to antibody secretion while biasing gene expression towards ribosomal translation, mitochondrial OXPHOS, and cell growth.

graph): ns, not significant; *P < 0.05; **P < 0.01; ***P < 0.001. **(D)** Multidimensional scaling (MDS) of differentially expressed genes at day 13 for the indicated samples. **(E)** Parsimonious Gene Correlation Network Analysis defined modules of differentially coregulated genes at day 13 shown as a heat map of module-level gene expression with expression patterns represented as the median z-score of the 10 most connected genes per module scale (−1.5 blue to +1.5 red). Conditions are identified as colour-coded blocks indicated above the figure, with samples from different donors identified in the blue to orange colour code. **(D, E)** Data are representative of two independent experiments with a total of n = 3 samples per time point and condition.

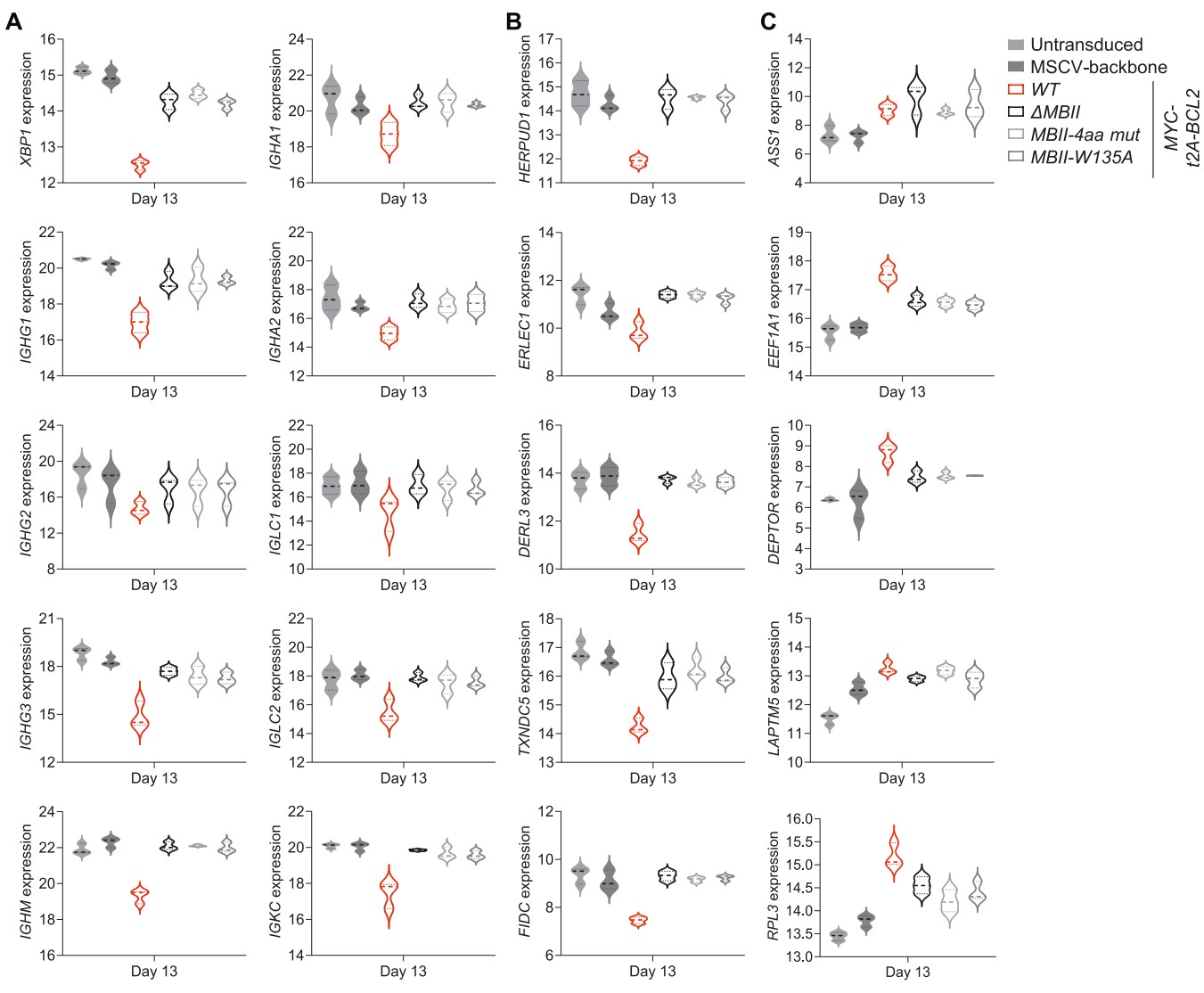

**Figure 8. Point mutation of the DCMW motif and W135 phenocopies MBII deletion.**
(A, B, C) Violin plots of log$_2$-normalized RNA-seq expression values of genes identified on the y-axis of each graph at day 13 for the conditions as indicated to the top right of the figure: (A) *XBP1* and immunoglobulin genes; (B) XBP1 targets; and (C) MYC targets. Data are representative of two independent experiments with a total of n = 3 samples per time point and condition. FDR-corrected *P*-values for all pairwise conditions at each time point are provided separately in Table S2.

The MYC TAD harbours evolutionarily conserved amino acid sequences essential for MYC-mediated transformation ([15], [20], [21]). Our data demonstrate a similar dependence on MB domains for the transcriptional impact of MYC during B-cell differentiation. In this setting, ΔMBI resulted in relatively enhanced MYC protein expression similar to T58I and consistent with ablation of the phosphodegron sequence ([23]). ΔMB0 led to a partial suppression of *XBP1* and secretory reprogramming and a reduced ability to induce a select subset of genes positively regulated by MYCwt. In contrast, ΔMBII or selective amino acid mutations, including the single point substitution W135A, rendered MYC all but non-functional. The MBII domain is particularly notable for recruitment of TRRAP and associated histone acetyltransferase complexes ([20], [26], [27]). The DCMW motif and W135 sit at the heart of the predicted

interface with TRRAP ([30]). Nevertheless, the degree to which MBII deletion, mutation of DCMW, or W135 impacted was unexpected. MBII deletion and related mutations reduced the absolute level of MYC overexpression observed relative to the MYCwt, MYC-T58I, or ΔMBI. We therefore do not exclude that part of the explanation for the decreased impact of MBII deletion or mutation lies in protein destabilization. However, the detectable levels of MYC observed for the ΔMBII deletion or DCMW/AAAA and W135A mutations remained significantly higher than expression of endogenous MYC. Therefore, reduced MYC expression levels are unlikely to provide a sufficient explanation for the profound effect of the MBII deletion and point mutations, and alternate explanations are suggested by well-established data in other systems. Although only a very small proportion (0.04%) of MYC-regulated

genes remained differentially expressed in the ΔMBII deletion or DCMW/AAAA and W135A mutations, interestingly this included *BCL2L11*, encoding BIM, which has long been identified as a critical factor limiting MYC-driven transformation through apoptotic induction ([35], [37]). This suggests that this pro-apoptotic function of MYC overexpression may be retained in the absence of MBII function, although this would be mitigated in the model system through accompanying BCL2 expression.

The MBII domain is critical for MYC-mediated transformation in fibroblasts, and a W135E substitution phenocopied the impact of MBII deletion ([28]). Interestingly in this cellular context, the W135A mutation had a limited impact on MYC function. MBII mutants can bind to physiological MYC targets in U2OS cells ([29]). Furthermore, ΔMBII MYC could partially compensate for MYCwt in *Drosophila* development ([88]). That all three versions of MYC targeting MBII in our model produce the same effect provides evidence for a critical dependence on MBII in the context of MYC deregulation during PC differentiation. Other studies have identified the interaction of MYC with WDR5 via the MBIIIb region of the MYC TAD as critical for recruitment of MYC to chromatin, including in Burkitt lymphoma ([89], [90]). The MBIIIb region along with the MYC DNA binding domain is intact in ΔMBII and MBII mutants. However, it remains to be tested whether the profound impact on overexpressed MYC in B-cell differentiation is explained by a requirement for MBII to support recruitment of excess MYC to target genes or to drive subsequent gene regulation. A recent study has shown that in U2OS cells, high-level MYC stabilizes and extends long-range chromatin interactions ([91]). If MYC DNA binding is retained in the MBII mutants, a possible explanation for the global impact of MBII mutants could be an essential role of MBII in mediating such chromatin effects.

In summary, we have tested a model that allows the acute effect of MYC overexpression in cooperation with BCL2 to be studied as human B cells differentiate to the PC state. MYC overexpression drove an aberrant cell state in relation to metabolic and cell growth–related features. Although promoting an aberrant phenotypic state, MYC overexpression left the elements of differentiation associated with repression of the B-cell state and functional differentiation to an ASC state intact. The TAD domains MB0 and MBII were necessary for overexpressed MYC effects, and a critical dependence on MBII could be resolved to the DCMW motif and W135.

# Materials and Methods

## Peripheral blood donors and cell lines

Peripheral blood was obtained from leukocyte cones of healthy anonymous donors (NHSBT) or by blood collection from healthy volunteers following the appropriate consent requirements. HEK-293 cells were maintained in culture using DMEM (41965039; Thermo Fisher Scientific) with 10% heat-inactivated foetal bovine serum (HIFBS) and 1:100 penicillin/streptomycin (15140122; Gibco) (DMEM CM). Irradiated murine fibroblasts transfected with human CD40L-L (CD40L-L stromal cells) were cultured in IMDM (31980-022;

Gibco) with 10% HIFBS (IMDM CM). Cells were incubated at 37°C with 5% $CO_2$.

## Plasmids and cloning

The retroviral construct *T58I-t2A-BCL2*, containing human *MYC* sequence, with T58I substitution, in combination with *BCL2* and a CD2 reporter, was kindly provided by the Daniel Hodson group, as well as the packaging and envelope plasmids pHIT60 and GALV-MTR, respectively ([41], [42]). A *WT-t2A-BCL2* construct containing the *MYC* WT sequence in combination with *BCL2* was also designed and synthesized commercially. To produce the TAD MB deletion mutants, DNA fragments of the MB0, MBI, and MBII MYC domains corresponding to 5′-YDSVQPYFYCDEEENFY-3′, 5′-PSE-DIWKKFELLPTPPLSP-3′, and 5′-IIIQDCMWSGFSAAAK-3′, respectively, were designed to be deleted from the *WT-t2A-BCL2* insert. The MYC TAD MB deletion mutants, as well as the mutation of MYC 132-135 amino acid sequence, DCMW, to alanine residues, and the single substitution W135A were commercially synthesized with the *t2A-BCL2* sequence included in the insert and cloned into the pIRES2-EGFP vector (Clontech). All the *MYC* mutants containing *t2A-BCL2* were subcloned into the MSCV-IRES-huCD2 plasmid kindly provided by the Daniel Hodson group. Plasmid propagation and successful ligations took place using NEB Stable Competent *E. coli* (C3040; NEB) for all the MSCV-based constructs and DH5α *E. coli* (18265-017; Invitrogen) for pIRES2-EGFP plasmids following the manufacturer's instructions, respectively. Diagnostic restriction enzyme digests verified all the constructs, and additional Sanger sequencing validated the MYC 132-135 amino acid mutation into alanine and the W135A mutant. Plasmids were stored at −20°C.

## Human memory B-cell differentiation system

PBMCs were isolated by Lymphoprep density gradient, at 800*g* centrifugation for 20 min at RT. PBMCs were washed with PBS, counted, and labelled appropriately, based on a human memory B-cell isolation kit (130-093-546; Miltenyi Biotec), with B-cell biotin antibody cocktail for 20 min at 4°C. Magnetic isolation of total B cells was performed upon incubation for 20 min at 4°C with anti-biotin beads. Following the previously established in vitro plasma cell differentiation method ([46]), memory B cells were isolated after magnetic labelling with anti-CD23 beads (130-094-510; Miltenyi Biotec) and cocultured in a 24-well plate format at 2 × 10$^5$ cells/ml with 2 × 10$^4$ cells/ml CD40L-L stromal cells in IMDM CM containing IL-2 (40 U/ml), IL-21 (100 ng/ml), and F(ab')2 fragments goat anti-human IgG, IgM, and IgA (20 μg/ml) (109-006-064; Jackson ImmunoResearch). At day 3 of B-cell differentiation in vitro, activated B cells were seeded without CD40L-L stromal cells in a new 24-well plate at 0.33 × 10$^5$ cells/ml for the *T58I-t2A-BCL2*, *WT-t2A-BCL2*, and *ΔMBI-t2A-BCL2* conditions and at 1 × 10$^5$ cells/ml for the rest of the conditions, in IMDM CM containing IL-2 (20 U/ml), IL-21 (50 ng/ml), and supplements (lipid mixture 1; chemically defined [200X] and MEM amino acid solution [50X]). At day 6, the *T58I-t2A-BCL2*, *WT-t2A-BCL2*, and *ΔMBI-t2A-BCL2* cells were re-seeded at 0.66 × 10$^5$ cells/ml, and the rest of the conditions at 2 × 10$^6$ cells/ml in IMDM CM containing APRIL (100 ng/ml), IL-21 (10 ng/ml), IL-6

(10 ng/ml), and supplements at 1 ml final volume per well. At day 9, the *T58I-t2A-BCL2*, *WT-t2A-BCL2*, and *ΔMBI-t2A-BCL2* cells were split following a 1:2 ratio, and appropriate volume of the day 6 complete medium was added to all the conditions aiming at a final volume of 2 ml per well. At day 13 onwards, the *T58I-t2A-BCL2*, *WT-t2A-BCL2*, and *ΔMBI-t2A-BCL2* cells were re-seeded at $0.33 \times 10^5$ cells/ml, and the rest of the conditions at $1 \times 10^6$ cells/ml in IMDM CM containing APRIL (100 ng/ml), IL-6 (10 ng/ml), and supplements at 2 ml final volume per well.

### Retroviral production and viral stock validation

HEK-293 cells seeded in 10-cm Petri dishes 24 h in advance were transfected with 1 ml Opti-MEM (31985062; Invitrogen) mixed with 18 $\mu$l Transit-293T (MIR 2700; Mirus) transfection reagent containing 1 $\mu$g pHIT60 packaging, 1 $\mu$g GALV-MTR envelope, and 4 $\mu$g retroviral constructs, as previously described ([41], [42]). The virus was collected and filtered after 48 h of incubation at 37°C with 5% $CO_2$ and either used fresh for transductions or aliquoted at 1 ml and stored at −80°C. To validate the frozen viral stocks, HEK-293 cells were seeded in six-well plates 24 h in advance and transduced with 1 ml frozen virus per well mixed with 10 $\mu$g/ml polybrene (sc134220; INSIGHT Biotechnology). A spinfection step at 1,250$g$ for 60 min at 30°C was used to augment retroviral infection, and the medium was replaced with fresh DMEM CM. CD2 staining and flow cytometry assessment were conducted 72 h post-transduction.

### Retroviral transduction of human memory B cells

Human memory B cells cocultured with CD40L-L were centrifuged at 390$g$ for 4 min at RT. Subsequently, 80% of the growth medium was aspirated and the cocultured activated memory B cells were transduced with 1 ml fresh or frozen retrovirus mixed with 25 $\mu$M HEPES (15630-056; Thermo Fisher Scientific) and 10 $\mu$g/ml polybrene (sc134220; INSIGHT Biotechnology). A spinfection step at 1,460$g$ for 90 min at 32°C was used to augment retroviral infection, and 70% of the medium was replaced with fresh IMDM CM containing IL-2 (20 U/ml) and IL-21 (50 ng/ml) ([41], [44], [45], [46]).

### Flow cytometry

Cells were washed with RT PBS and centrifuged at 450$g$ for 5 min at RT. Live/dead fixable viability stain 780 nm (565388; BD Biosciences) was diluted 1:1,000 in RT PBS; 500 $\mu$l was shared per sample and incubated for 15 min at RT. Cells were washed with 2 ml FACS buffer (PBS + 0.5% HIFBS) and centrifuged at 450$g$ for 5 min at RT. 25 $\mu$l blocking buffer containing hIgG molecules (I2511-10MG; Sigma-Aldrich) and natal mouse serum in FACS buffer were added, and samples were incubated for 15 min at RT. 10 $\mu$l antibody master mix was added followed by 20 min at RT incubation. Stained cells were washed with 2 ml FACS buffer and centrifuged at 450$g$ for 5 min at RT. Cells were fixed with 150 $\mu$l 2% paraformaldehyde and stored at 4°C for flow cytometry analysis. For Ki67/EdU, intracellular staining was performed by permeabilizing the cells in saponin-based permeabilization buffer for 20 min and staining was followed for 1, 5 h at 4°C. For transcription factors, intracellular

staining was performed using Triton X-100 (PBS/0.2% Triton X-100) permeabilization. Antibodies used were as follows: CD19-PE (130-113-169; Miltenyi), CD20-ef450 (48-0209-42; Invitrogen), CD20-BV421 (562873; BD Biosciences), CD27-FITC (555440; BD Pharmingen), CD38-PE-Cy7 (335825; BD Biosciences), CD138-APC (130-117-395; Miltenyi), CD2-BUV395 (563820; BD Biosciences), Ki67-Alexa Fluor 488 nm (558616; BD Biosciences). For intracellular stains with transcription factors, CD19-PE was replaced with PE-conjugated primary antibodies to IRF4 (56649; BD Pharmingen), BLIMP1 (564702; BD Pharmingen), and XBP1 (NBP1-77681PE; Novus) with serum and/or isotype controls (for IRF4—554680; BD Pharmingen; for BLIMP1—554689; BD Pharmingen; for XBP1—NBP2-24983). CountBright beads (C36950; Invitrogen) were used for absolute cell number analysis in combination with trypan blue–based haemocytometer counts. Flow cytometry data were collected using CytoFLEX S and CytoFLEX LX analysers (Beckman Coulter). Flow cytometry analysis was conducted using FlowJo software v.10.7.2 and v.10.8.1 and GraphPad Prism 10 software.

### Proliferation assays

5-Ethynyl-2′-deoxyuridine (EdU) 1-h pulse assay took place for the day 21 in vitro differentiated untransduced and *T58I-t2A-BCL2* cells. EdU incorporation was detected using the Click-iT Plus EdU Alexa Fluor 647 kit (C10635; Invitrogen) based on the provided protocol and followed by Ki67 intracellular staining as described above. Cellular proliferation was assessed by flow cytometry.

### Western blotting

Cells were washed with sterile PBS, and protein was extracted with 30 $\mu$l RIPA buffer. Lysed cells were kept on ice for 15 min and centrifuged at 13,500$g$ for 30 min at 4°C after supernatant collection. BCA assay (AR0146; Boster, or 23250; Thermo Fisher Scientific) was performed for protein quantification according to the manufacturer's instructions. Normalized lysates were loaded in SDS–PAGE followed by wet-Western blotting for the detection of c-MYC (D3N8F rabbit, 1:1,000 13987S; Cell Signaling Technology), BLIMP1 (PRDM1a from mouse, 1:1,000), BCL2 (2870S, 1:1,000; Cell Signaling Technology), BCL2 (2,872, 1:1,000; Cell Signaling Technology), and $\beta$-actin (A1978-200UL, 1:10,000; mouse). HRP-conjugated rabbit or mouse secondary antibodies were used at 1:10,000. Membrane development was performed using enhanced chemiluminescent HRP substrate (34580; Thermo Fisher Scientific) incubation, and images were taken with ChemiDoc MP Imaging System (Bio-Rad). Membranes were stripped using stripping buffer (46430; Thermo Fisher Scientific) for 10 min at RT.

### ELISA and ELISpot

Human IgG (A80-104A; Bethyl) and human IgM (A80-100A; Bethyl) ELISA quantification sets were used according to the manufacturer's instructions for total IgG and IgM antibody secretion detection, respectively. Standard curves were generated from readings of absorbance at 450 nm with a Cytation 5 imaging plate reader (BioTek). Analysis was conducted using MyAssays Ltd online data analysis tool and GraphPad Prism 10 software.

ELISpot was performed across a range of seeding cell densities, according to standard protocols using Millipore 0.45 $\mu$m plates (MAIPSWU10) and Mabtech IgM kit (3880-2H) and IgG kit (3850-2H) with TMB substrate (3651-10; Mabtech). Plates were analysed using CTL Immunospot S6 Ultra-V Analyser (S6ULTRA-02-6121) and CTL ImmunoSpot SC Suite.

### RNA extraction

Cells were counted and lysed with 800 $\mu$l to 1 ml TRIzol (15596026; Ambion) reagent, incubated for 10 min at RT, and stored at –80°C. Chloroform (C2432-26ML; Honeywell) was added at 160 $\mu$l or 200 $\mu$l, respectively, to defrosted samples followed by vigorous shaking for 15 s and 3 min at RT incubation. A centrifugation step at 11,400$g$ for 15 min at 4°C was followed by collection of the aqueous phase. 400 $\mu$l isopropanol and 10 $\mu$l glycogen (AM9510; Invitrogen) were added, and the mixture was incubated for 10 min at RT. Samples were centrifuged at 11,400$g$ for 10 min at 4°C, and the pellet was washed three times with 75% ethanol followed by a centrifugation step at 7,600$g$ for 5 min at 4°C. RNA pellets were air-dried for 10 min and dissolved in 30 $\mu$l RNase-free water. RNA was incubated at 55°C for 10 min, treated with DNases for genomic DNA removal (AM1906; Invitrogen), and stored at –20°C.

### RNA-seq analysis

RNA-seq was conducted on a NovaSeq 6000 platform (Illumina), using 150-bp paired-end sequencing. The fastq files were assessed for initial quality using FastQC v0.11.8, trimmed for adapter sequences using TrimGalore v0.6.10, and aligned to GRCh38.p13/hg38 with STAR aligner (v2.6.0c) (92). Transcripts were re-annotated with the MyGene.info API using all available references and any ambiguous mappings manually assigned. Transcript abundance was estimated in RSEM v1.3.1 and imported into R v4.1.2 with txImport v1.22.0 and then processed using DESeq2 v1.34.0 (93, 94, 95, 96). Software DESeq2 determined differential gene expression (DEG) between every contrast and a total DEG carried out with a likelihood ratio test (LRT), quality was visualized using MA plots, and shrinkage of log fold was estimated using the apeglm method (Table S1) (97). Log$_2$-transformed expression values were normalized and stabilized with variance-stabilizing transformation (VST).

### RNA-seq network

The PGCNA approach was used (for details and validation of the PGCNA approach, see our other work) (73). The transcripts differentially expressed between any contrast or across the time series data (DESeq2 FDR < 0.01) were retained for PGCNA. 15,941 genes giving a 15,941 × 32 matrix were analysed with PGCNA in Fig 3. The equivalent numbers corresponding to Figs 4C and 7E were 14,360 genes giving a 14,360 × 48 matrix and 7,148 genes resulting in a 7,148 × 18 matrix, respectively. These were used for a PGCNA2 (–n 1,000, –b 100) giving a network with multiple modules. The median expression per condition/time was visualized as Z-scores mapped onto the network. For each gene in the network, a strength (edge-weight × degree) was calculated and used to select the top 10 genes per module. These were converted to module expression values (MEVs) by taking their median z-scores (across samples) and visualized as a hierarchically clustered heat map.

### Enrichment analysis

The gene signature enrichment (GSE) was assessed using a hypergeometric test, in which the draw is the gene list genes, the successes are the signature genes, and the population is the genes present on the platform. The resultant $P$-values are then adjusted for multiple testing using the Benjamini and Hochberg correction. For the PGCNA network GSE analyses, the genes per module were compared against the 43,572-signature database (background: 15,941 or 14,360 or 7,148 genes in the network for RNA-sequencing data analysis corresponding to Figs 3, 4C, and 7E, respectively). Only signatures that contain at least three genes in the background set were retained.

### Data processing and availability

All RNA-seq data analyses were undertaken on ARC4, part of the High-Performance Computing Facilities at the University of Leeds, UK. Interactive networks and all metadata are available at https://matthewcare.wixsite.com/pgcna/myc-trd. PGCNA python scripts are available at https://github.com/medmaca/PGCNA.

### Statistical analysis

Statistical tests, in non–RNA-seq-derived data, were performed using one-way ANOVA or an unpaired two-tailed $t$ test with GraphPad Prism 10 software.

# Data Availability

The primary datasets are available at the Gene Expression Omnibus GSE262809.

### Ethics statement

Approval for this study was provided by UK National Research Ethics Service via the Leeds East Research Ethics Committee, approval reference: 07/Q1206/47, IRAS reference 187050.

# Supplementary Information

# Acknowledgements

We thank Erica Wilson for technical support and advice and Ulf Klein and Richard Bayliss for advice, support, and critical review of this work. This work was supported by the Ella Dickinson Charitable Foundation Scholarship

Program (scholarship recipients: P Vardaka, B Kemp, and A Annahar; investigators: R Owen, GM Doody, and RM Tooze) and an Intercalated Degree 1022 Award, Pathological Society of Great Britain and Northern Ireland and EXSEL scholarship, University of Leeds School of Medicine, to E Page. This work was supported by Cancer Research UK Programme Grant (C7845/A29212) (to MA Care, GM Doody, and RM Tooze) and Cancer Research UK and FC AECC and AIRC under the Accelerator Award Program (C355/A26819). This work was supported by Blood Cancer UK project grant 23021 (to S Stephenson, GM Doody, and RM Tooze). This research is funded in part (M.C.) by the National Institute for Health and Care Research (NIHR) Leeds Biomedical Research Centre (BRC) (NIHR203331). RM Tooze and GM Doody are supported in part by the National Institute for Health and Care Research (NIHR) Leeds Biomedical Research Centre (BRC) (NIHR203331). DJ Hodson was supported by a fellowship from Cancer Research UK (CRUK) (RCCFEL\100072) and received core funding from Wellcome (203151/Z/16/Z) to the Wellcome-MRC Cambridge Stem Cell Institute and from the CRUK Cambridge Centre (A25117). DJ Hodson is supported by the National Institute for Health and Care Research (NIHR) Cambridge Biomedical Research Centre (BRC-1215-20014). The views expressed are those of the authors and not necessarily those of the NIHR or the Department of Health and Social Care. For the purpose of Open Access, the authors have applied a CC BY public copyright licence to any Author Accepted Manuscript version arising from this submission.

## Author Contributions

P Vardaka: data curation, formal analysis, investigation, visualization, and writing—original draft, review, and editing.
B Kemp: formal analysis, investigation, and visualization.
S Stephenson: data curation, formal analysis, investigation, visualization, and writing—review and editing.
E Page: formal analysis and investigation.
MA Care: resources, data curation, software, formal analysis, visualization, and writing—review and editing.
M Umpierrez: investigation.
A Annahar: formal analysis and investigation.
E O'Callaghan: investigation.
R Owen: conceptualization, funding acquisition, and project administration.
DJ Hodson: conceptualization, resources, and methodology.
GM Doody: conceptualization, formal analysis, supervision, funding acquisition, visualization, project administration, and writing—review and editing.
RM Tooze: conceptualization, supervision, funding acquisition, project administration, and writing—original draft, review, and editing.

## Conflict of Interest Statement

The authors declare that they have no conflict of interest.

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
