## [Reviewer comments · Life Science Alliance]

Life Science Alliance

Enforced MYC expression directs a distinct transcriptional state during plasma cell differentiation

Panagiota Vardaka, Ben Kemp, Sophie Stephenson, Eden Page, Matthew A Care, Michelle Umpierrez, Adam Annahar, Eleanor O'Callaghan, Roger Owen, Daniel J Hodson, Gina M Doody and Reuben M Tooze

DOI: <https://doi.org/10.26508/lsa.202402814>

Corresponding author(s): Prof. Reuben M Tooze (University of Leeds)

Review Timeline:

Submission Date:	2024-05-09
Editorial Decision:	2024-07-03
Revision Received:	2025-06-03
Editorial Decision:	2025-07-01
Revision Received:	2025-07-07
Accepted:	2025-07-08

Scientific Editor: Tim Fessenden

Transaction Report:

July 3, 2024

Re: Life Science Alliance manuscript #LSA-2024-02814-T

Prof. Reuben M Tooze
University of Leeds
Leeds Institute of Medical Research
St James's University Hospital
Beckett Street
Leeds LS9 7TF
United Kingdom

Dear Dr. Tooze,

Thank you for submitting your manuscript entitled "Enforced MYC expression selectively redirects transcriptional programs during human plasma cell differentiation" to Life Science Alliance. The manuscript was assessed by expert reviewers, whose comments are appended to this letter. We invite you to submit a revised manuscript addressing the Reviewer comments.

Thank you for this interesting contribution to Life Science Alliance. We are looking forward to receiving your revised manuscript.

Sincerely,

B. MANUSCRIPT ORGANIZATION AND FORMATTING:

Reviewer #1 (Comments to the Authors (Required)):

LSA-2024-02814-T

The authors examined the effects of MYC and BCL2 overexpression on human plasma cell (PC) differentiation in vitro. This is relevant because MYC and BCL2 overexpression occur in a poor-outcome group called double expressors and understanding how MYC and BCL2 may interfere with B cell differentiation is therefore relevant. Early in the cell culture process, human B cells were transfected with a retroviral construct overexpressing MYC and BCL2. This led to increased cell size and numbers, improved maintenance of the B cell-like phenotype, and a significant reduction in cells displaying PC markers by Day 13. MYC-BCL2 also seemed to reduced antibody secretion per cell, as determined by ELISA (please see comments below). RNA sequencing revealed that BLIMP1, IRF4, PAX5 and EBF1 are all slightly reduced suggesting an in B-plasma cell intermediate. The expression of XBP1 is significantly impaired, however, how this happens is mechanistically unclear. These phenotypes are reverted if mutations are introduced in the MBII domain, and more specifically in the DCMW motif and W135 residue. These regions of MYC are known to interact with TRRAP, however, if the block in plasma cell differentiation is due to this protein interaction is not explored. In conclusion although the manuscript investigates an interesting question there are significant concerns that require addressing.

Figure 1

- Although the authors show CD2 expression as a readout of effective viral infection, the authors need to demonstrate OE of BCL2 and MYC from the construct.
- In figure 1F the statistical value is comparing what to what? T581-t2A-BCL2 versus both controls or only untransduced? E.g., respecting the % of CD19 positive cells at Day20, it seems that the results would not be as statistically significant if T581-t2A-BCL2 is compared to MSCV-backbone.
- For the measurement of IgG and IgM (Figure 1G) statistics are not provided.
- The work would benefit substantially if cells would be also characterized by intracellular stain for the expression of BLIMP1 and IRF4 together with surface markers. This would provide a better molecular definition and if the OE of MYC and BCL2 interferes with the expression of these transcription factors at the protein level.
- The measurement of antibody production per cell may be flawed. To measure the number of cells producing antibody the authors need to perform ELISPOT at each specific timepoint. The amount of antibody measured at e.g., day 13 may reflect a cumulation of antibody produced also in previous timepoints.
- Mechanistically, how do the authors consider XBP1 is suppressed?

Figure 2

- For RNAseq analysis - were live cells isolated? Given the massive cell death observed in the cell culture this step would seem wise (see Fig. S1C).
- Figure 2 - Again, the work would add significantly benefited if the authors would have performed intracellular stain for some of the factors represented in the manuscript by gene expression. This is particularly relevant because there are modest reductions at the gene expression level for PAX5, BLIMP1 and IRF4. Similarly, it would be relevant to demonstrate that the more profoundly suppressed induction of XBP1 is reflected at the protein level.
- Figure 2 - Statistics need to be calculated in the data presented.

Figure 4

- What is the cell count in the various MYC mutant conditions, e.g., was proliferation equally maintained across the various MYC variants? Cells counts must be available because of data presented in Figure 5E.
- According to Figure S3C the levels of MYC attained vary widely across the various MYC mutants, how is therefore possible to compare results? In contrast BCL2 is relatively constant which suggests that the various mutants have possibly different stability, and this seems to be particularly the case for those mutants lacking MB0 and MBII which are those showing the phenotype reversion.
- Possibly as consequence and according to PCGNA analysis MBII mutant seems to no longer activate MYC target genes (M1) and the MB0 mutant also seems impaired. How do the authors interpret these results? Is it possible that the authors observe a

reverted phenotype because the mutants have impaired MYC activity? If yes, is this a complete null or lack of specific components of the program.

- Similarly to Figure 1, antibody measurement shown in Figure 5E should be performed using ELISPOT to clearly determine how many cells produce antibody and the amount of antibody production per cell, rather than an averaging out the amount of antibody per cell.

Figure 6

- Similar questions as those asked for Figure 5 are asked for Figure 6. Are the MYC mutants inducing identical proliferation, cell expansion as compared to WT?
- Do the authors consider that the mutants are MYC nulls as suggested in Figure 6E module 2 enrichment? Or do the authors consider that activity of MYC is largely maintained but specifically lacking the interaction with TRRAP? If the latter the authors should perform MYC immunoprecipitation to demonstrate lack of TRRAP interaction with MYC. Is there a change in TRRAP-Dependent MYC target genes? An inhibitor for MYC-TRRAP interaction seems to exist PMID: 34676047 and could be used for experiments? Further, it would be important to perform MYC-CHIP-seq to determine changes in MYC-DNA binding patterns.
- Same questions as before respecting antibody measurement.
- The authors finish the manuscript with the following statement "We conclude that either a four amino acid DCMW/AAAA substitution or a single W135A substitution sufficed to phenocopy deletion of MBII and abrogate features associated with MYC overexpression in human B-cell differentiation. ". If this is true than other features of MYC activity are maintained independently of the mutants, can the authors describe which features are maintained and which are lost?

Reviewer #2 (Comments to the Authors (Required)):

In this manuscript by Vardaka et al., the authors use an in vitro retroviral overexpression approach to assess how Myc and Bcl2 deregulation impact human B cell differentiation. They find that Myc and Bcl2 deregulation leads to loss of plasma cell surface phenotype and impaired secretory output. Myc overexpression did not result in B cell transformation. The ability of Myc to regulate B cell differentiation and function is dependent on the Myc homology box MBII.

Deregulation of the transcription factor Myc is frequently found in individuals with B cell lymphoma. Here the authors sought to contribute to understanding of how Myc drive B cell lymphoma through acute overexpression of the Myc variant T58I. Mutations in T58 are often found in aggressive lymphomas. The authors perform extensive flow cytometric and RNA-Seq analysis at multiple time points following in vitro activation to assess phenotypical and functional differences in the B cell response. They then determine key functional domains required for Myc to limit secretory reprogramming. Limitations of the study include a lack of assessment of the role of Myc using an in vivo system and the relative incremental advance of this study beyond what was already known from past work. However, the high quality of the analysis performed by the author's and their clear presentation of their findings enable this paper to still represent a valuable contribution to the field.

Major issues:

1) It is difficult to understand the main takeaways from the PGCNA and GSEA analysis in figure 3 from looking at the figure. This is partially due to the large number of groups and the small size of much of the text. The authors should try to find an alternative way to present this data that makes it easier for the reader to grasp the key conclusions. For example the author may want to separately plot some of the most important results and move the complete analysis to the supplemental figures.

Minor issues:

1) Several of the figures (Fig. 3a, 4e, 6d) contain text that is blurry. The authors should rewrite this text to make it easier for the reader to understand.

We thank the reviewers for their valuable comments relating to our manuscript entitled “Enforced MYC expression selectively redirects transcriptional programs during human plasma cell differentiation”.

To address the key points, we have performed a new set of differentiations with MYC wild type and MB deletion mutants and assessed phenotypes alongside intracellular flow cytometry and ELISpot analyses. This necessitated set up of new flow approaches, and there were delays in supply of antibodies (for example PAX5 flow cytometry antibodies remained on back order for extreme amounts of time and in the end could not be used due to this). Furthermore, the laboratory has relatively limited staff and resources balancing the required experiments with their primary roles. We do not have the resources available to repeat all the constructs/virus combinations suggested by reviewer 1. Hence, we focused on what we considered to be the key point around MYCwt and MB deletions.

These results are presented in new Figure 6 and demonstrate two key things: 1) expression of BLIMP1 and IRF4 at protein levels are not impacted by MYC over-expression and despite the induced aberrant phenotypes, these factors retain a relationship to the suppressed markers of plasma cell differentiation; 2) as assessed by ELISpot, over-expression of MYC or MB deletion mutants does not lead to a loss of ASC number.

We thank reviewer 1 for asking for this experiment, and in view of the result we have changed the conclusions of the manuscript around the impact of MYC on plasma cell differentiation. The results nonetheless demonstrate that MYC overexpression establishes an aberrant differentiation state combining aberrant phenotype, secretory activity and metabolic reprogramming, and that this impact depends on MBII and W135.

We have also focused on clarifying and simplifying the presentation of the modular gene expression data. Together we consider the manuscript is substantially improved, and the key issues of the reviewers are addressed to the extent that we are able.

Reviewer 1

Figure 1

- Although the authors show CD2 expression as a readout of effective viral infection, the authors need to demonstrate OE of BCL2 and MYC from the construct.

- *This is addressed in new Supplemental Figure 1b and for other constructs in Supplemental Figure 3c and 5c.*

- In figure 1F the statistical value is comparing what to what? T58I-t2A-BCL2 versus both controls or only untransduced? E.g., respecting the % of CD19 positive cells at Day20, it seems that the results would not be as statistically significant if T58I-t2A-BCL2 is compared to MSCV-backbone.

- *The differences were evaluated using ANOVA comparing the means across the three-group comparison using a fixed effect model.*

- For the measurement of IgG and IgM (Figure 1G) statistics are not provided.

- *In view of the new results generated from ELISpot as requested by the reviewer, the discussion of these data has changed. Lines 136-138 now describe the data (now Figure 1h) as demonstrating that “establishment of functional antibody secretion was evident for both IgG and IgM in T58I-t2A-BCL2 conditions by day 6 and sustained at day 13”.*

- The work would benefit substantially if cells would be also characterized by intracellular stain for the expression of BLIMP1 and IRF4 together with surface markers. This would provide a better molecular definition and if the OE of MYC and BCL2 interferes with the expression of these transcription factors at the protein level.
 - *This is addressed for MYCwt and deletion mutants in new Figure 6. The data confirm that IRF4 and BLIMP1 protein are equivalently expressed after enforced MYC expression. We were unable with resources available to perform intracellular flow cytometry across repeats of all transductions/constructs tested. We therefore selected to test this with the MYCwt and MB deletions as the most encompassing comparison in terms of MYC biology.*

- The measurement of antibody production per cell may be flawed. To measure the number of cells producing antibody the authors need to perform ELISPOT at each specific timepoint. The amount of antibody measured at e.g., day 13 may reflect a cumulation of antibody produced also in previous timepoints.
 - *We accept that the per cell secretion rate as previously represented has limitations/flaws. It does reflect a cumulative antibody secretion. We have removed these data and the derived conclusions from the manuscript. We have performed ELISpot analyses alongside the transcription factor intracellular flow as suggested. As noted in the comment above, we do not have the resources available to perform this testing across all constructs with sufficient replicates. Therefore, as with the transcription factor flow cytometry, we selected to address this point with a new set of n=4 differentiations with MYCwt and the deletion mutants combining intracellular assessment of TFs with phenotype and ELISpot assessments. This is included in new Figure 6. The ELISpot analyses demonstrate that an antibody secreting cell state is established in the context of enforced MYC over-expression. In view of this result the conclusion that MYC delays transition to an ASC state has been removed from the manuscript. New data are included in results under section heading “Differentiation to an antibody secreting cell state is retained with enforced expression of MYC” (line 292-313) and with changes in the abstract, discussion and summary Supplementary Figure.*

- Mechanistically, how do the authors consider XBP1 is suppressed?

We were technically unable to verify a loss of XBP1 protein expression due to limitations of the antibody and isotype combination used. The data (not shown) failed to show a significant difference in XBP1 expression if compared to serum rather than isotype staining. We have reconfigured the discussion of results in light of the new ELISpot data.

Since prior studies have shown that MYC can bind to the XBP1 promoter we reason that the effect of over-expressed MYC may be at the level of XBP1 promoter activity. This would require over-expressed MYC to act as a transcriptional repressor of XBP1 in the context of PC differentiation (line 404-409). Differences from previous studies in cancer cell lines might be explained by differential cofactor recruitment. Further studies would be needed to determine whether this is the mechanism. These are beyond the currently funded/supported capacity of the laboratory to address.

Overall, we have taken the reviewers criticisms on board and in view of the new experimental data have adjusted the conclusions reached. We note this does not impact on the fundamental conclusion related to dependence on MBII and W135.

Figure 2

- For RNAseq analysis - were live cells isolated? Given the massive cell death observed in the cell culture this step would seem wise (see Fig. S1C).

Total RNA was isolated, there was no live cell isolation step.

While there is cell death, our experience is that the cell debris does not contribute to the detected expression signal. This is presumably due to mRNA degradation.

As an example of evidence in support of this in our original description of the model system in which mRNA was equivalently isolated (Cocco et al 2012), we demonstrated that within the detected mRNA the XBP1 is acutely responsive to DTT mediated ER stress. Short term application of DTT drives almost complete splicing of XBP1u to XBP1s. Such a response is indicative of an intact ER/UPR response pathway. This could not derive from a dead cell.

Further evidence supporting the validity of the approach is found in the fact that the mRNA expression profiles derived from equivalent isolation procedures in plasmablasts/plasma cells, as extensively documented in our work, show time dependent patterns of expression change reflecting active responses to applied external factors and lead to progressive and biologically relevant gene programme responses that correspond with functional and phenotypic changes in plasma cells such as secretory outputs and surface phenotypes. Thus, the mRNA changes observed corresponds with features of live cells in the cultures.

Relating to the suggestion of live cell isolation: we do use viable cell enrichment steps in other applications of the method. However, following isolation recovery is needed to ensure that sensitive signalling responses do not lead to spurious variability in expression data. Thus, the possible benefits of removing cell debris are limited and are outweighed by the downsides in terms of gene expression changes associated with manipulations linked to enrichment.

Thus, we conclude that the approach used reflects expression features of viable cell populations. And we would argue that there is reasonable evidence to suggest that steps necessary for cellular isolation themselves come with significant risks of confounding effects.

- Figure 2 - Again, the work would add significantly benefited if the authors would have performed intracellular stain for some of the factors represented in the manuscript by gene expression. This is particularly relevant because there are modest reductions at the gene expression level for PAX5, BLIMP1 and IRF4. Similarly, it would be relevant to demonstrate that the more profoundly suppressed induction of XBP1 is reflected at the protein level.

As noted above we have addressed this point in relation to MYCwt and MB deletions and have established intracellular flow cytometric detection to do so in new Figure 6. This has been informative for BLIMP1 and IRF4. Due to reagent limitations assessment of XBP1 proved uninformative. There was an inordinate level of background with the isotype control giving a higher signal than XBP1 antibody itself and even in control XBP1s transfected populations the approach proved unconvincing. If isotype was replaced with serum control, there was no difference

between conditions for XBP1 expression levels. Since we could not demonstrate a reliable shift between isotype control and specific antibody and ELISpot results failed to support a decrease in ASC number we have changed the presentation of results and the conclusions reached as noted above.

- Figure 2 - Statistics need to be calculated in the data presented.
We have addressed this point across all the gene vignette examples (Figure 2, 5, 8 and Sup. Figure 2,4 and 6) by including a new supplemental Table (Sup. Table 2) in which all the FDR corrected p-values for all possible pairwise comparisons at each time point are provided. We have selected this approach because the figures were at risk of becoming overly crowded with the various significances indicated, and because as suggested in an earlier comment there are challenges with interpreting which comparison the significance indicator applies to. By including data in a supplemental table, we are providing a clear representation of each comparison and consider that this fully addresses the point of calculating and presenting the statistics for this and related figures.

Figure 4

- What is the cell count in the various MYC mutant conditions, e.g., was proliferation equally maintained across the various MYC variants? Cells counts must be available because of data presented in Figure 5E.

*The cell counts are now shown in new Figure 4b right panel – these demonstrate changes consistent with the impact of
MYCwt=MYC Δ MI> Δ MB0> Δ MBII=MSCV=untransduced*

- According to Figure S3C the levels of MYC attained vary widely across the various MYC mutants, how is therefore possible to compare results? In contrast BCL2 is relatively constant which suggests that the various mutants have possibly different stability, and this seems to be particularly the case for those mutants lacking MB0 and MBII which are those showing the phenotype reversion.

The reviewer is correct to point out that expression levels of MYC mutant protein levels vary more widely than those of BCL2. This includes both the relative increase in MYC observed with MBI deletion and a decrease in detected MYC levels with MBII and MB0 deletion. We have included a comment that the impact of these deletions on MYC function may relate both to changes in protein stability as well as dependence of the MB domain on co-factor recruitment/MYC function (lines 320-324) and lines (429-436) in the discussion.

Since the expression of BCL2 is constant we can conclude that transduction and expression from vectors was similar. The question we are addressing is whether MYC function is compromised by deletion of an MB domain, the experiments demonstrate that this is the case. The mechanism of this compromise may be multifactorial. It may both be via impacts on stability and on MYC function. We articulate this more explicitly now to address the point raised.

Given that the levels of MB0 and MBII deleted MYC are still substantially greater than those in control cells (Sup. Figure 3c and 5c) and there is minimal evidence of differential gene expression we can conclude that MBII deletion or mutation of

DCMW or W135 compromises the ability of over-expressed MYC to drive aberrant gene expression.

- Possibly as consequence and according to PCGNA analysis MBII mutant seems to no longer activate MYC target genes (M1) and the MB0 mutant also seems impaired. How do the authors interpret these results? Is it possible that the authors observe a reverted phenotype because the mutants have impaired MYC activity? If yes, is this a complete null or lack of specific components of the program.

We agree with the reviewer that MYC related function is compromised profoundly in the MBII mutant. The extent of this compromise approaches a complete loss of function. We do conclude that we see a loss of MYC activity in the context of MBII deletion.

While expression levels are reduced possibly related to stability, there is still significant retained MYC expression, and thus MYC functional activity does also appear significantly impaired. This is consistent with observations in other model systems, as we discuss.

The extent of compromise for MBII deletion or DCMW/AAAA or W135A is close to a complete null phenotype as we demonstrate in Figures 4, 5, 7 and 8 and related discussion (e.g. lines 340-347 and 436-440). This, we argue, provides evidence that this domain and specific residue are critical for over-expressed MYC to drive aberrant gene expression.

- Similarly to Figure 1, antibody measurement shown in Figure 5E should be performed using ELISPOT to clearly determine how many cells produce antibody and the amount of antibody production per cell, rather than an averaging out the amount of antibody per cell.
As discussed above ELISpot is addressed in new experimental data shown in new Figure 6. We have focused the experiment to address this point on the MYCwt and MB domain deletions. The data demonstrated that ASCs are generated similarly with MYC transduction. As noted in previous comments we have extensively modified the discussion and conclusion in relation to this point.

Figure 6

- Similar questions as those asked for Figure 5 are asked for Figure 6. Are the MYC mutants inducing identical proliferation, cell expansion as compared to WT?

Cell count data are now included as requested in Figure 7c new right panel. The changes are significant. Relative to MYCwt the MYC mutants are ineffective at inducing cell expansion, and the transduced populations show no significant difference from controls.

- Do the authors consider that the mutants are MYC nulls as suggested in Figure 6E module 2 enrichment? Or do the authors consider that activity of MYC is largely maintained but specifically lacking the interaction with TRRAP? If the latter the authors should perform MYC immunoprecipitation to demonstrate lack of TRRAP interaction with MYC. Is there a change in TRRAP-Dependent MYC target genes? An inhibitor for MYC-TRRAP interaction seems to

exist PMID: 34676047 and could be used for experiments? Further, it would be important to perform MYC-CHIP-seq to determine changes in MYC-DNA binding patterns.

We apologise for the lack of clarity. In terms of gene expression and phenotypic impact the mutants abrogate MYC function. We have avoided the use of the “null” term given that the experiments are addressing an exogenous over-expression model. As we discuss the reasons for this are likely multifactorial, a contributing factor may be lack of TRRAP activity. The citation PMID:34676047 to our reading describes a luminescence assay for the potential detection of inhibitors of the MBII:TRRAP interaction and validates this using MBII domain mutations (we note including the W135A mutation), the manuscript does not as far as we can tell describe an inhibitor as suggested. There has been a recent manuscript from another group describing a green tea derivative compound (EGCG) which may structurally interfere with MYC TAD folding. We respectfully rebut the suggestion that an exploration of the dependence on TRRAP or other interactions including ChIPseq and inhibitor studies are necessary to support the conclusions of a role for MYC MBII in aberrant gene regulation in our model. Such experiments are the subject of an independent project which we are developing for funding submission.

- Same questions as before respecting antibody measurement.

We have addressed this point in relation to the MB domain deletions. We have not been able given the available resources to additionally test the MBII mutants. Since the MBII deletion and MBII mutants are very similar in phenotype we consider this should sufficiently address this point.

- The authors finish the manuscript with the following statement "We conclude that either a four amino acid DCMW/AAAA substitution or a single W135A substitution sufficed to phenocopy deletion of MBII and abrogate features associated with MYC overexpression in human B-cell differentiation.". If this is true than other features of MYC activity are maintained independently of the mutants, can the authors describe which features are maintained and which are lost?

To clarify, in the phrase “abrogate features associated with MYC overexpression”, we were using the word “abrogate” in the sense of “do away with”. To clarify this, we have changed the wording to the following: "We conclude that either a four amino acid DCMW/AAAA substitution or a single W135A substitution sufficed to phenocopy deletion of MBII and abolish the vast majority of features associated with MYC overexpression in human B-cell differentiation."

In relation to the question: “describe which features are maintained and which are lost?” We have included additional clarification of this with all significance values for pairwise comparisons included in tab 3 of new Supplemental Table 2. We have also described this more explicitly on lines 339-347. Essentially 99.6% of MYC enforced gene expression changes are lost when considering gene expression changes that are consistent across pairwise comparisons for MBII deletion or the two point mutations against controls. The residual genes that remain differentially expressed are now mentioned “DEPTOR, EEF1A1, KISS1R, BCL2L11, EEF2, EVI2A, GAS5, RPL5 TGFBI, ZNF581 were upregulated while XBP1 remained significantly reduced in expression level”. Of note this includes BCL2L11 encoding BIM which is a classical MYC target

gene implicated in control of MYC driven oncogenic transformation. We have included discussion of this on lines 436-440.

Reviewer 2

1) It is difficult to understand the main takeaways from the PGCNA and GSEA analysis in figure 3 from looking at the figure. This is partially due to the large number of groups and the small size of much of the text. The authors should try to find an alternative way to present this data that makes it easier for the reader to grasp the key conclusions. For example the author may want to separately plot some of the most important results and move the complete analysis to the supplemental figures.

We have taken the reviewers suggestion on board and have simplified and refined all the expression data representations in new Figure 3, Figure 4c and Figure 7e. We have also rewritten the description of these data to clarify further and align with the new Figure 3 on lines 210-235.

We have removed the gene signature enrichment analysis representation as this figure was in fact redundant with the tabulated versions of the data in Supplemental Tables 4, 7 and 10.

Minor issues:

1) Several of the figures (*Fig. 3a, 4e, 6d*) contain text that is blurry. The authors should rewrite this text to make it easier for the reader to understand.

We apologise for this issue and have included new representations which should have addressed this issue.

July 1, 2025

RE: Life Science Alliance Manuscript #LSA-2024-02814-TR

Prof. Reuben M Tooze
University of Leeds
Leeds Institute of Medical Research
St James's University Hospital
Beckett Street
Leeds LS9 7TF
United Kingdom

Dear Dr. Tooze,

Thank you for submitting your revised manuscript entitled "Enforced MYC expression redirects transcriptional programs during human plasma cell differentiation". As you will see, reviewers are overall satisfied with the revisions in place. The remaining requests from Reviewer 1 are left to your discretion. We appreciate their consideration of the main conclusions of this work and how these are encapsulated by the current title. To our view, redirecting transcriptional programs during differentiation seems consistent with the revised conclusion of "a distinct aberrant expression state during PC differentiation." However you are welcome to revise the title of this work if you wish. We would be happy to publish your paper in Life Science Alliance pending final revisions necessary to meet our formatting guidelines below.

- Please be sure that the authorship listing and order is correct.
- Please add the X and Bluesky handles of your host institute/organization as well as your own or/and one of the authors in our system.
- There is a name discrepancy for two of the co-authors, please correct: Adam Annahar in the manuscript file vs. Adam Mabbutt in the system; Eleanor O'Callaghan in the manuscript file vs. Elleanor O'Callaghan in the system.
- Please add an Author Contributions section to your main manuscript text.
- If Figure S7 represents the Graphical Abstract, please change the file designation to Graphical Abstract and remove its legend from the manuscript text as well as the figure call-outs.
- Please add call-out for Figure S6B and S6C to your main manuscript text.
- Please add molecular weight markers for the blots in Figures S1, S3, and S5.

A. FINAL FILES:

-- Summary blurb (enter in submission system): A short text summarizing in a single sentence the study (max. 200 characters including spaces). This text is used in conjunction with the titles of papers, hence should be informative and complementary to

the title. It should describe the context and significance of the findings for a general readership; it should be written in the present tense and refer to the work in the third person. Author names should not be mentioned.

B. MANUSCRIPT ORGANIZATION AND FORMATTING:

Sincerely,

Reviewer #1 (Comments to the Authors (Required)):

Overall, the authors have addressed this reviewer's major concerns with new experimental data and appropriate adjustments to their conclusions. However, two important points remain that should be addressed prior to publication:

1. The authors state that they used ANOVA with a fixed effect model to compare the three experimental conditions (T58I-t2A-BCL2, MSCV-backbone, and untransduced) shown in Figure 1F. While ANOVA tests for overall differences among groups, it does not identify which specific groups differ. No post-hoc pairwise comparisons (e.g., T58I vs MSCV) are presented, which limits the interpretability of the results, particularly given that the MSCV control appears more similar to the T58I group than to the untransduced group in some stains.

2. Based on additional data provided in response to earlier comments, the authors now conclude that enforced MYC expression does not fundamentally interfere with plasma cell differentiation. In the discussion (line 383), they state: "Rather, overexpressed MYC established a distinct aberrant expression state during PC differentiation." In light of this, the manuscript title should be revised to reflect this refined conclusion. Specifically, the phrase "redirects transcriptional programs during human plasma cell differentiation" may mislead readers into thinking MYC disrupts the differentiation process itself, which the data no longer support.

Reviewer #2 (Comments to the Authors (Required)):

The authors have adequately addressed my concerns.

July 8, 2025

RE: Life Science Alliance Manuscript #LSA-2024-02814-TRR

Prof. Reuben M Tooze
University of Leeds
Leeds Institute of Medical Research
St James's University Hospital
Beckett Street
Leeds LS9 7TF
United Kingdom

Dear Dr. Tooze,

Thank you for submitting your Research Article entitled "Enforced MYC expression directs a distinct transcriptional state during plasma cell differentiation". It is a pleasure to let you know that your manuscript is now accepted for publication in Life Science Alliance. Congratulations on this interesting work.

DISTRIBUTION OF MATERIALS:

Again, congratulations on a very nice paper. I hope you found the review process to be constructive and are pleased with how the manuscript was handled editorially. We look forward to future exciting submissions from your lab.

Sincerely,
